# On the Mechanism of the Ionizing Radiation-Induced Degradation and Recycling of Cellulose

**DOI:** 10.3390/polym15234483

**Published:** 2023-11-22

**Authors:** Richard List, Lorelis Gonzalez-Lopez, Aiysha Ashfaq, Amira Zaouak, Mark Driscoll, Mohamad Al-Sheikhly

**Affiliations:** 1Department of Chemical Engineering, State University of New York College of Environmental Science and Forestry, Syracuse, NY 13210, USA; 2UV/EB Technology Center, State University of New York College of Environmental Science and Forestry, Syracuse, NY 13210, USA; 3Department of Materials Science and Engineering, University of Maryland, College Park, MD 20742, USA; 4Department of Chemistry and Biochemistry, University of Maryland, College Park, MD 20742, USA; 5Research Laboratory on Energy and Matter for Nuclear Science Development, National Center for Nuclear Science and Technology, Sidi-Thabet 2020, Tunisia; amirazaouak@gmail.com; 6Department of Chemistry, State University of New York College of Environmental Science and Forestry, Syracuse, NY 13210, USA

**Keywords:** cellulose ionizing-radiation, degradation, recycling, structure

## Abstract

The use of ionizing radiation offers a boundless range of applications for polymer scientists, from inducing crosslinking and/or degradation to grafting a wide variety of monomers onto polymeric chains. This review in particular aims to introduce the field of ionizing radiation as it relates to the degradation and recycling of cellulose and its derivatives. The review discusses the main mechanisms of the radiolytic sessions of the cellulose molecules in the presence and absence of water. During the radiolysis of cellulose, in the absence of water, the primary and secondary electrons from the electron beam, and the photoelectric, Compton effect electrons from gamma radiolysis attack the glycosidic bonds (C-O-C) on the backbone of the cellulose chains. This radiation-induced session results in the formation of alkoxyl radicals and C-centered radicals. In the presence of water, the radiolytically produced hydroxyl radicals (^●^OH) will abstract hydrogen atoms, leading to the formation of C-centered radicals, which undergo various reactions leading to the backbone session of the cellulose. Based on the structures of the radiolytically produced free radicals in presence and absence of water, covalent grafting of vinyl monomers on the cellulose backbone is inconceivable.

## 1. Introduction

Cellulose is the most abundant biopolymer on Earth [1]. It has a tremendous global economic importance, being produced at a rate of 1.5 × 10^12^ tons per year [2,3]. It can be obtained from a vast number of sources, such as woody and herbaceous plants, algae, tunicates, colorless protists, as well as photosynthetic and heterotrophic bacteria [4,5,6,7,8,9]. Cellulose is a linear homo-polysaccharide composed of repeating β-D-glucopyranose units linked by a β-1,4 glycosidic bond [10]. As each glucopyranose group contains three free hydroxyl groups, there is extensive hydrogen bonding between individual polymer strands [5,11]. Hydrogen bonding along with van der Waals forces facilitate a parallel stacking of cellulose molecules into nanofibers, which further assemble into cellulose microfibrils [4,12]. The resulting hierarchical order imparts cellulose microfibrils with excellent mechanical strength, allowing them to act as reinforcing components in the natural architectures of organisms [13]. In human history, cellulose-based materials have been widely used for thousands of years as fundamental engineering materials, paper, construction, energy, textile, and furniture [14]. At present, interdisciplinary cellulose research and product development has expanded the application of cellulose into all aspects of human life. Cellulose applications include esters and ethers for coatings, films, membranes, building materials, pharmaceuticals, biofuels, and other biomaterials [13]. Cellulose is an abundant, renewable, and highly versatile resource whose products can be readily recycled while also being biodegradable [15]. This review provides an extensive survey on cellulose and its derivatives, focusing on the development of novel materials utilizing ionizing beam technologies.

Ionizing Radiation is defined by the International Union of Pure and Applied Chemistry as: “Any radiation consisting of directly or indirectly ionizing particles or a mixture of both, or photons with energy higher than the energy of photons of ultraviolet light or a mixture of both such particles and photons” [16]. Ionizing radiation encompasses various particles, such as alpha particles, beta particles, gamma rays, X-rays, neutrons, high-speed electrons, protons, and other particles capable of generating ions. As this radiation traverses a substance, it releases sufficient energy to create ions by rupturing molecular bonds and displacing electrons from atoms or molecules [17]. 

### 1.1. Cellulose Sources

Wood-based cellulose has been extracted from both softwoods, such as spruce and pine, and hardwoods, such as maple and oak [18]. Softwood fibers contain 45–50% cellulose, 18–35% hemicellulose, and 23–35% lignin. Hardwood fibers contain 40–50% cellulose, 24–40% hemicellulose, and 18–25% lignin [19]. The degree of polymerization of cellulose from macerated wood is 6000–10,000 DP, while from pulped wood fibers it is 2000–4000 [20]. Wood fibers have an elastic modulus of 14–40 GPa, a tensile strength of 380–1240 MPa, and an elongation to rupture of 3–22% [10].

Plant-based cellulose has been extracted from a broad diverse range of plant biomass material [18]. Most plant fibers contain 30–75% cellulose, 10–35% hemicellulose, and 0–2% lignin. Cotton fibers do not contain lignin or hemicellulose [19]. The degree of polymerization of cellulose for most plant biomass is 1000–2600 DP. Cotton fibers range from 10,000–15,000 DP [20]. Plant fibers have an elastic modulus of 5–130 GPa, a tensile strength of 300–1050 MPa, and an elongation to rupture of 1–8% [10].

Bacterial-based cellulose was identified by Adrian J. Brown in a jelly-like membrane on the surface of a vinegar fermentation broth in 1886 [21]. It is made at terminal complexes on the surface of bacteria. Excreted at greater than 90% purity, bacterial cellulose does not require extensive chemical treatment [22]. The degree of polymerization of cellulose for most bacteria-based cellulose is 7000–16,000 DP. They have an elastic modulus of 60–115 GPa [10].

Algal-based cellulose has been isolated from Grey, Green, Red, and Brown algae [18]. Algal cellulose is the primary component in algal cell walls in fibers that contain hemicellulose, proteins, and lignin [23]. The degree of polymerization of cellulose for algal biomass is 2500–4300 DP [20].

Tunicate-based cellulose is isolated from invertebrate animals known as Ascidiacea (sea squirts), which are made up of over 2300 species [24]. Tunicate-based cellulose is derived from the outer tissue of tunicate or “tunic”, from which an untainted form of cellulose called “tunicin” can be isolated [25]. Tunicate-based cellulose has a degree of polymerization of between 700 and 3500 DP [20].

### 1.2. Cellulose Structure

Cellulose is an unbranched, natural polymer composed of repeating glucose units (C_6_H_10_O_5_)_n_ [26], and is considered as the most profuse organic material and polysaccharide on Earth. The chemical structure of cellulose consists of glucose units joined together by glycosidic oxygen bonds. The chemical structure of cellulose is made of β-(1,4)-linked glucopyranose units forming a high molecular weight (MW) linear homopolymer. Each glucopyranose unit is oriented 180° with respect to its neighbors [27]. The repeating unit is, therefore, a dimer titled cellobiose. The degree of polymerization of cellulose chains varies depending on the source [3].

Cellulose is typically found in nature in the form of microfibrils in the cell walls of wood, plant, and algae tissues, and the membrane of epidermal cells of tunicates. It can also be synthesized by bacteria which generate nanofiber networks. Cellulose, when synthesized, forms long individual chains that eventually come together in a hierarchical manner at the site of biosynthesis. These chains assemble into elementary fibrils, also known as protofibrils, which have an approximate diameter of 3.5 nm [28]. The diameter of these elementary fibrils can vary between 2 and 20 nm, depending on the source of cellulose, and their packing arrangements are influenced by the conditions of biosynthesis [29]. Microfibrils are formed through the aggregation of elementary fibrils. This aggregation occurs due to various forces, including van der Waals forces, as well as intra and intermolecular hydrogen bonds. These forces lead to the coalescence of elementary fibrils, reducing the free energy of the surfaces involved [30]. It is important to note that the cellulose molecules aggregated to form microfibrils can have different orientations depending on the source material. The number of chains grouped within a fibril bundle varies depending upon its source [31]. The microfibril aggregates are not homogenous. Some sections of the chain consist of tightly packed cellulose chains that are held together with strong hydrogen bonds forming crystallites. Other areas have chains that are much less ordered, forming amorphous regions [32,33]. Microfibril bundle structures span in size from nanoscale to macroscopic dimensions, with a cross-sectional dimension that varies depending on the source of synthesis [18,34]. The degree of crystallinity found in a cellulosic biomass depends on its origin, extraction method, and pretreatment. The degree of crystallinity of wood-based and plant-based cellulose usually ranges from 40 to 60%, while cellulose from bacteria and tunicin range from 80 to 100% [10,22,35].

Six polymorphs of crystalline cellulose have been identified (I, II, III_I_, III_II_, IV_I_, IV_II_). The structure of each polymorph varies depending on the source of cellulose and the conditions from which it was harvested and isolated [36]. Native cellulose generally has the crystal structure of cellulose I, that is subdivided into allomorphs Iα and Iβ [10,37]. The ratio of Iα to Iβ structures depends on the source of cellulose. The Iα structure with a triclinic unit cell is the allomorph known for most algal and bacterial cellulose [38]. Cellulose forming cell walls of higher plants is in the cellulose allomorph Iβ. These microfibrils assemble into larger units called fibrils, and they in turn further assemble to form the cell wall [28]. Cellulose I can be converted to cellulose II and is typically obtained by regeneration (dissolution and recrystallization) or mercerization (aqueous sodium hydroxide treatment) of native cellulose [38,39]. During this conversion, the parallel chain arrangement of cellulose I changes into cellulose II.

Through a liquid ammonia treatment, cellulose III can be formed from cellulose I or II, and this product is known as cellulose III_I_ and III_II_, respectively. Cellulose IV is formed through thermal treatments of cellulose III_I_ and III_II_ [38,40].

### 1.3. Cellulose Extraction

Isolation and recovery of cellulose from a biomass source requires the removal of lignin, hemicellulose, and other biological components. The production of cellulose on an industrial scale includes thermal pulping, mechanical pulping, acid hydrolysis, bisulfate hydrolysis, and kraft pulping (NaOH and Na_2_S) [41]. On a smaller scale, maceration is used to preserve the physical structure and properties of cellulose from a specific biomass source. Maceration is a chemical process that dissolves portions of the biomass to release substructures for analysis [42]. Mild macerations act upon the middle lamella, yielding intact cell wall structures and macrofibers. More aggressive macerations act on lignin and hemicellulose, yielding cellulose fibers [43]. Many procedures have been attempted and cataloged [44,45].

### 1.4. Morphological Forms of Cellulose

Cellulose morphological forms are classified by size, morphology, aspect ratio, crystallinity, and physiochemical properties. Cellulose morphological forms are classified as follows.

### 1.5. Cellulose Fibers

Cellulose, synthesized as long individual chains, coalesce in a series of steps forming hierarchical assemblies of elementary fibrils (protofibrils), at the site of biosynthesis with an approximate diameter of 3.5 nm [28]. Elementary fibrils are then clustered into cellulosic fibers that when processed produce fibers in three typical geometries, strand fibers (long fibers of 20–100 cm length), staple fibers (short fibers of 60 mm length), and pulp fibers (very short fibers of 1–10 mm length) [46].

### 1.6. Cellulose Filaments

Cellulose can be reassembled from solution into filament form by wet-extrusion [47], hydrodynamic flow-focusing [48], or spinning [49]. These processes produce filaments with different physicochemical aspects that are generally strong and ductile [50].

### 1.7. Cellulose Nanofibrils

Cellulose nanofibrils are produced through mechanical, chemical, and/or enzymatic processing of cellulosic biomass [13,29,34]. These processes break the cellulose polymer at the amorphous regions, producing particles with “fuzzy” ends made up of individual strands of cellulose [29]. The width of CNFs ranges from 10 to 100 nm depending on the source of cellulose, fibrillation process, and pretreatment [18].

### 1.8. Crystalline Cellulose

Cellulose fibers can be processed to yield only the crystalline region of the cellulose polymer, forming cellulose nanocrystals (CNCs) [12,27,37]. Large-scale production of CNCs is achieved through sulfuric acid hydrolysis and the ultrasonic treatment of bulk cellulose that results in the breakdown of the polymer and the release of CNC particles upon extraction [18]. CNCs exhibit excellent characteristics, such as a high aspect ratio, large specific strength modulus, and surface. They are also abundant, biodegradable, and possess reactive surfaces that give them the ability to make stable suspensions [29].

### 1.9. Regenerated Cellulose

Regenerated cellulose is produced by physical dissolution, treatment of the cellulose solution to remove contaminants or to add functionality, and regeneration with the addition of an anti-solvent. The regeneration process generally transforms cellulose into the cellulose II polymorph [51]. The traditional method for producing cellulose fibers is known as the viscose process. It begins by treating cellulose with sodium hydroxide, followed by the derivatization of cellulose using carbon disulfide, resulting in a highly viscous solution called sodium xanthogenate. Subsequently, this solution undergoes treatment with an acidic solution to regenerate the cellulose [52]. Solvent systems used to dissolve and then regenerate cellulose are outlined below.

### 1.10. N-Methylmorpholine-N-Oxide (NMMO)

NMMO has been extensively studied as an organic solvent in the creation of cellulose solutions [51]. This solvent, unlike derivatizing solvents, does not chemically modify cellulose, but rather facilitates a purely physical dissolution process. This is achieved through the disruption of the hydrogen bonding networks within cellulose, leading to the formation of solvent complexes characterized by hydrogen bonds between the cellulose macromolecules and NMMO [53].

The crystal structure in the cellulose/NMMO mixture collapses between 70 and 100 °C to become mobile and transforms crystalline cellulose into the amorphous [54].

### 1.11. Ionic Liquids (ILs) 

Ionic liquids (ILs) are compounds containing ions that remain in a liquid state below 100 °C. They possess several appealing properties, including non-volatility, chemical and thermal stability, and non-flammability [55]. The choice of IL constituents allows for control over cellulose solubility and solution properties. The dissolution of cellulose can be significantly accelerated through the application of microwave heating. The elevated chloride concentration in ILs disrupts the hydrogen bond networks within cellulose. However, the presence of water diminishes cellulose solubility by competitively forming hydrogen bonds with cellulose microfibrils [56]. To precipitate cellulose from solution, water, ethanol, or acetone can be added. The resulting regenerated cellulose exhibits a similar degree of polymerization and polydispersity as the original cellulose, but its morphology undergoes significant changes, with fused microfibrils forming a relatively uniform macrostructure [55]. As the ionic solution is easily recovered, the dissolution of cellulose through the use of ionic liquids has provided new opportunities for “green” cellulose utilization [57]. Wang (2012) [52] reviewed over sixty ionic liquids being used to process cellulose.

### 1.12. LiCl/N,N-Dimethylacetamide (LiCl/DMAc)

A commonly used solvent system for dissolving various types of cellulose without causing significant degradation is LiCl/DMAc. This solvent system is non-derivatizing and non-aqueous in nature. It finds wide application in cellulose analysis techniques like size-exclusion chromatography, nuclear magnetic resonance, and light scattering [51]. The typical method for dissolving cellulose using LiCl/DMAc involves a series of steps. First, the cellulose undergoes mechanical disintegration and activation. Then, a solvent exchange process is carried out, starting with swelling the cellulose in water, followed by subsequent swelling in ethanol, methanol, or acetone. Finally, the cellulose is dissolved in LiCl/DMAc to complete the process [58]. This approach ensures successful dissolution of cellulose in LiCl/DMAc while minimizing the risk of degradation.

### 1.13. NaOH Aqueous Solutions

Cellulose exposed to hydroxide solutions exhibits various behaviors, including disintegration into rod-like fragments upon dissolution, swelling followed by complete fiber dissolution, swelling leading to partial fiber dissolution, and uniform swelling without any fiber dissolution [59,60]. A mechanism for these observations involves the rate at which a solvent can penetrate weak amorphous regions causing expansion or ballooning. The resultant stress on the fiber matrix will cause fragmentation. The solvent will quickly diffuse and/or dissolve the small cellulose chain fragments [61].

### 1.14. Alkali/Urea and NaOH/Thiourea Aqueous Solution

The solubility cellulose in NaOH/urea and NaOH/thiourea is dependent on temperature, crystallinity, and molecular weight [62,63]. 

Cellulose pulps with molecular weights below 114,000 can be dissolved within 2 min in a solution containing 7 wt% NaOH and 12 wt% urea. Similarly, cellulose pulps with molecular weights lower than 372,000 can be dissolved within the same time frame in a pre-cooled (−12 °C) aqueous solution containing 4.6 wt% LiOH and 15 wt% urea [64]. The low-temperature dissolution process of cellulose involves the attachment of NaOH to cellulose chains through the creation of hydrogen-bonded networks while the urea hydrates as a shell surrounding the NaOH hydrogen-bonded cellulose to form a complex leading to the dissolution of cellulose [64].

### 1.15. Tetra Butyl Ammonium Fluoride/Dimethyl Sulfoxide System

Without any pretreatment, cellulose can be dissolved in dimethyl sulfoxide (DMSO) at room temperature within 15 min when it is mixed with tetrabutylammonium fluoride trihydrate (TBAF·3H_2_O) in concentrations ranging from 10 to 20% (*w*/*v*) [65].

### 1.16. Metal Complex Solutions

Cellulose dissolution, characterization, and regeneration are facilitated by the use of aqueous metal complex solutions containing transition metal ions and nitrous ligands. The most widely recognized examples of such solutions are cuprammonium hydroxide and cupriethylenediamine hydroxide [51].

### 1.17. Molten Inorganic Salt Hydrates

Molten inorganic salt hydrates are also effective in cellulose dissolution. According to Heinze et al., and Yang et al., some of the widely used cellulose-dissolving inorganic molten salt hydrates include ZnCl_2_·4H_2_O, Zn(NO_3_)_2_·XH_2_O (X < 6), FeCl_3_·6H_2_O, LiClO_4_·3H_2_O, LiI·2H_2_O, LiBr, LiSCN·2H_2_O, ZnCl_2_·3H_2_O, Ca(SCN)_2_·3H_2_O, as well as mixtures such as LiClO_4_·3H_2_O/MgCl_2_·6H_2_O, LiClO_4_·3H_2_O/Mg(ClO_4_)_2_/H_2_O, LiCOI_4_·3H_2_O/NaClO_4_/H_2_O, LiCl/ZnCl_2_/H_2_O, NaSCN/KSCN/LiSCN/H_2_O, NaSCN/KSCN/Ca(SCN)_2_/H_2_O [66,67].

### 1.18. Functionalization of Cellulose

The utility of cellulose can be expanded by functionalization of its structure, producing a wide variety of derivatives. The characteristics of cellulose derivatives are determined by their degree and type of substitution, as well as the pattern of functional groups along the polymer chain. The primary focus of modification is the abundant hydroxyl groups present in cellulose. However, due to their low reactivity, harsh reaction conditions are often necessary to introduce additional functionalities [68]. Esterification of hydroxyl groups is the most common derivatization of cellulose. The derivatives produce diverse characteristics including chemical structure, moisture sorption, water interaction, surface activity, and solubility [18].

### 1.19. Cellulose Ethers

Cellulose hydroxyl groups can be partially or totally etherified by epoxides, alpha halogenated carboxylic acids, and halogenoalkanes [69]. Cellulose ethers dissolve slowly in acetic conditions and rapidly in alkaline conditions. The solubility of cellulose ethers depends on the chemical structure and the degree and pattern of substitution. Most water-soluble cellulose ethers have a degree of substitution of 0.4–2 [18,69]. Cellulose ethers include:Methyl cellulose—Methyl cellulose is the main commercial cellulose ether. It is the simplest alkyl ether and is synthesized in an alkaline medium with methylating agents like dimethyl sulfate or methyl chloride [70];Carboxymethyl cellulose—Carboxymethyl cellulose is synthesized by the reaction of cellulose with monochloroacetic acid. The C2, C3, and C6 hydroxyl groups are substituted by carboxymethyl groups. The substitution at the C2 position is favored slightly. There are no secondary OH groups formed in this process [71];Ethyl cellulose—The synthesis of ethyl cellulose involves the reaction between alkali cellulose and ethyl chloride, typically carried out at a temperature of approximately 60 °C for several hours [72,73];Hydroxyethyl cellulose—Hydroxyethyl cellulose, which is prepared from the reaction of alkali cellulose and ethylene oxide. The chemical structure of hydroxyethyl cellulose can be easily further modified due to its reactive hydroxyl groups [51];Hydroxypropyl cellulose—Hydroxypropyl cellulose is formed in a reaction with 1,2-propylene oxide. The secondary OH groups can further react to form other novel compounds [74].

### 1.20. Cellulose Esters

Cellulose esters are commercially used as a thermoplastic biopolymer. They have good solubility in common solvents and melt before decomposition [75]. Cellulose esters include:
Cellulose acetate—Cellulose acetate is produced through the esterification of cellulosic biomass [76]. It is relatively cheap since it is commonly obtained from waste products such as paper and agricultural byproducts [77];Cellulose nitrate, also referred to as nitrocellulose or celluloid, holds the distinction of being the inaugural semi-synthetic polymer in the realm of plastics. This substance is produced by subjecting cellulose to a chemical reaction with nitric acid. Within its structure, cellulose nitrate exhibits a polynitrate ester configuration, typically containing 2.2–2.8 nitrate groups per glucose unit [78];Cellulose sulfate—The sulfation of cellulose is done by the utilization of acids such as chlorosulfonic acid, sulfuric acid, or even sulfur trioxide [79].

### 1.21. Application of Cellulose and Its Derivatives

The tunability of the cellulose structure, and thus of its physical and chemical properties, makes it a leading material for advanced technologies [80]. Due to the high number of possible combinations of the previously discussed parameters (i.e., source, structure, and dissolution system) will produce cellulose materials with a large variety of properties, the applications are endless. One of the current focuses of cellulose research applications is on the skin-care industry [81,82,83]. Deposition of chitin nanofibrils on the skin through an electrospray method has been found to lead to anti-inflammatory activity. Azimi et al. found that the source and dissolution system chosen did not have any negative effects on the results [81]. Recently, interesting biological activity has been found in carboxymethyl cellulose (CMC) [82]. CMC is commonly used as a structural element. However, it has been found that it has inflammation control properties and has been presented as a possible replacement for surfactants used in cosmetics, which makes it an ideal alternative for skin-care applications [84,85]. 

Optoelectronic applications of cellulose have also been increasing in this last decade [86]. For example, the use of cellulose as high-definition displays for flexible touch screen panels is being explored. Zhu et al. fabricated a highly transparent cellulose-based paper that could be used in high-definition displays [87]. The possibility of fabricating these transparent films also opens the door for using cellulose in other flexible electronic applications. Chen et al. studied the fabrication of transparent and hydrophobic cellulose films, with not only an optical transparency of up to 92.3%, but also a tensile strength of 198.7 MPa [88]. These have promising applications for both electronic device protection and emerging electronics.

### 1.22. Principles of Ionization Radiation

The interaction of ionization radiation and matter has been reviewed extensively [89,90,91,92,93,94,95] and discussed in textbooks related to Electron Microscopy [96,97,98,99,100]. Before delving into the effects of ionizing radiation on cellulose, this section will give an overview of the effects of ionizing radiation on matter. Since polysaccharides have various conformations, the radiation effects will depend on the participation of specific monomers present in the cellulosic chains [101]. When cellulose is irradiated, free radicals are formed and interact with the solvent present, in the same way as synthetic polymers undergo free radical reactions. 

The interactions between ionizing radiation and organic material occur in a three-step process [91].
A.Physical stage (Ionization and excitation):

The physical stage occurs within 10^–13^ s after exposure to ionization radiation when electrons are knocked out of the target material, as a result of which positive ions are created in this material. The ionization potential for electrons in most organic molecules is in the range of 10 to 15 eV. The applied ionization radiation contains much more energy. Deposited energy that is not absorbed in ionization causes electronic excitation [91].
(1)$$\mathrm{Ionization}:\;\mathrm{AB}\to \mathrm{Radiation}\to \left[A\sim B\right]\to {\mathrm{AB}}^{+}+e^{-}$$
(2)$$\mathrm{Excitation}:\;\mathrm{AB}\to \left[A\sim B\right]\to \mathrm{Radiation}\to \mathrm{AB}*$$

The symbol of *EB* is electron beam energy, $\left[A\sim B\right]$ is used to designate electronic excitation, AB is an organic molecule, AB^+^ is a positive ion, *e*^−^ an electron, AB* excited molecules.
B.Physico-chemical stage (Free radical formation):

The second stage involves the formation of free radicals and occurs between 10^−13^ to 10^−6^ s irradiation. In this stage, abstracted electrons from the irradiated molecules are attracted back to the positive charged ions. Charge recombination occurs frequently during irradiation. The ionization potential (10 to 15 eV) is recovered, producing highly excited molecules. The recovered ionization potential is greater than the normal molecular bond strength of 3 to 4 eV, causing bonds to break into free radicals [91].Radical formation: AB* → A^●^ + B^●^(3)

AB* are excited molecules, A^●^ and B^●^ are uncharged fragments called free radicals.

In solid polymers, ions and free radicals can be trapped for a significant amount of time [91].
C.Chemical stage (Modifications in organic materials):

During the timeframe of 10^−6^ to 10^−1^ s following irradiation, the concluding phase occurs, leading to molecular structure alterations and the creation of organic materials exhibiting modified characteristics.

### 1.23. Reaction of Organic Free Radicals

Free radical chain reactions play a vital role in altering organic materials, as they provide the opportunity for a significant product yield with a minimal input of radiation energy. Ionizing radiation possesses enough energy to disrupt chemical bonds within organic materials, including polymers. The typical outcome is the breaking of chemical bonds and the creation of free radicals and ions. Ionization radiation produces radicals that initiate chain reactions producing desired outcomes. Typical chain reactions occurring with free radicals include crosslinking, polymerization, scissioning or breaking up of molecules, curing and grafting. In addition, the free-radical reactions are also responsible for other reactions including unsaturation, gas evolution, and oxidation. 

#### 1.23.1. Crosslinking and Polymerization

Crosslinking is the process of forming transverse covalent bonds between polymer chains to form a three-dimensional structure. Crosslinking is a reaction where bonds between carbon and hydrogen in neighboring polymer chains are broken by radiation and two polymeric radicals R^●^ are produced. Radicals combine to form an intermolecular bond or crosslink as shown below:RH → R^●^ + H^●^(4)
where RH is a hydrogen-containing polymer, R^●^ a polymeric radical, and H^●^ a hydrogen atom. 

This step is followed by hydrogen atom abstraction:H^●^ + RH → R^●^ + H_2_(5)
R^●^ + R^●^ → R − R (6)

In solid polymers, crosslinking exhibits high yields as it involves the mobility of radicals facilitated by hydrogen hopping, where hydrogen atoms transfer from one location to another [91,102]. This phenomenon enables the formation of new crosslinks, resulting in the release of hydrogen molecules. Unsaturated compounds, especially vinyl monomers, undergo efficient polymerization when exposed to ionizing radiation. During this process, the double bonds of monomers are broken, which leads to a chain elongation reaction with the formation of a polymer.

#### 1.23.2. Scissioning

Chain scission is the breaking of the chemical bonds of polymer chains, resulting in degradation of the polymer, e.g., cellulose. This leads to a decrease in the length and molecular weight of the macromolecule. Radicals formed in polymer materials with side branches transfer to hydrogen atoms forming a double bond; in this case, the scission becomes permanent. Figure 1 illustrates the process of permanent main-chain scission [103].

#### 1.23.3. Crosslinking versus Main-Chain Scission

Crosslinking plays a crucial role in polymer irradiation as it typically enhances mechanical and thermal properties, although the dominant effect—whether crosslinking or chain scission degradation—depends on various factors such as material properties, state, and chemical structure [102]. Materials with G_(S)_/G_(X)_ ratios below 1.0 are favored for crosslinking applications. G_(S)_ is the yield for scission, and G_(X)_ is the yield for crosslinking [103].

#### 1.23.4. Curing

Radiation curing is a process that involves the rapid conversion of a reactive liquid monomer into a solid through a combination of polymerization and crosslinking. This process offers several significant benefits, including the reduction or elimination of volatile organic compounds (VOCs). Furthermore, radiation curing is faster compared to chemical curing methods. Among the various types of radiation used for this process, electron beam radiation is particularly well-suited for radiation curing applications. Figure 2 illustrates a schematic representation of the radiation curing process [91]. 

Electron accelerators operating between 0.3 and 0.5 MeV are the optimal radiation sources for achieving deep penetration of the electron beam in curing processes. The curing process involves the generation of free radicals, which can be hindered by the presence of oxygen. To mitigate the inhibiting effect of oxygen, it is recommended to carry out irradiation in a vacuum or inert gas atmosphere and to cover the oligomer-monomer formulation with a polymer film during the irradiation process. By doing so, the negative impact of oxygen can be suppressed.

#### 1.23.5. Grafting

Graft copolymerization includes two simultaneous processes. Active sites formed on the surface of an existing polymer by irradiation and a monomer will then bond to these sites. Graft copolymerization of the added monomer will produce homo polymerization of the monomer [102].

The grafting degree is determined by calculating the percentage of mass increase in the polymer after the grafting process. Ionizing radiation will produce radical, ionic, and mixed radical—ionic mechanisms. The trunk polymer is activated and will initiate graft-polymerization of an added monomer. The grafted branches are attached to the trunk polymer by covalent bonds (Figure 3). Radiation is effective in activating polymers because polymeric free radicals are produced by irradiation and initiate polymerization [103].

#### 1.23.6. Unsaturation 

Intra-chain double bonds are formed through scission of branched polymers as follows:R′-CH_2_-CH_2_-R″ → R′-CH=CH-R″ + H_2_(7)

The presence of chlorine in polymers, such as polyvinylchloride (PVC), leads to the formation of double bonds more efficiently, but also resulting in the generation of HCl through a chain reaction. This can pose challenges as it causes significant corrosion on metal components situated near irradiated PVC. When polymers containing double bonds are exposed to radiation, their unsaturation level tends to decrease. The introduction of free radicals onto these double bonds encourages the formation of crosslinks [90].

#### 1.23.7. Gas Evolution

Gases are produced in an irradiation process of polymers. Hydrogen is the primary gas generated by irradiation and is formed because of a two-step process. A carbon–hydrogen bond is broken, followed with the elimination of a hydrogen atom radical H^●^ in accordance with reactions (5) and (7). The yield of hydrogen evolution in polyethylene irradiation is a molar ratio of 4 to 1. Polypropylene and polyisobutylene will produce methane as the short side branches are selectively removed. Carbon monoxide and carbon dioxide are released from polyacrylates and polymethacrylates. Acidic gases such as HCl and HF are produced from halogenated polymers [7].

#### 1.23.8. Influence of Atmospheric Oxygen

Oxygen contains two unpaired electrons that will react with a free radical, R^●^, generating peroxyl radicals RO_2_^●^ as
R^●^ + O_2_ → RO_2_^●^
(8)

Peroxyl radicals have a strong affinity for hydrogen atoms and abstract them from neighboring RH molecules, resulting in new radicals, R^●^.
RO_2_^●^ + RH → RO_2_H + R^●^
(9)

This chain reaction will oxidize organic molecules. RO_2_H is unstable and slowly breaks down at room temperature, releasing radicals causing oxidative damage. 

### 1.24. Radiation-Chemical Reactions in Aqueous Solutions

Comprehending the radiation chemistry of water holds significant importance, especially in relation to bioprocessing [102]. Water is often present in biomass materials and can undergo irradiation in various forms such as aqueous solutions, suspensions, or pastes. When high-energy ionizing radiation passes through water or a diluted aqueous solution, it generates electrons, positively charged water ions, and excited water molecules.
(10)$$H_{2}O\overset{\;\mathrm{Radiation}\;}{\to }e^{-1},H_{2}O^{*},\;H_{2}O^{●+}$$

The ejected electron loses energy through ionization and excitation events and eventually becomes thermalized and solvated.
*e^−^* + nH_2_O → *e*_aq_^−^(11)

The positive radical ions, H_2_O^●+^, are thought to be energetically very unstable and to decompose in 10^−13^ s, giving H^+^ ions and HO^●^ radicals.
H_2_O^●+^ → H^+^ + HO^●^(12)
H_2_O^●+^ + H_2_O → H_3_O^+^ + HO^●^

H_2_O* represents an excited water molecule which may ionize, dissociate, or simply return to the ground state.
H_2_O* → H^+^ + HO• or
H_2_O* → *e^−^* + H_2_O^●+^(13)

The primary species either react with one another or diffuse into the solution. Additional hydrogen atoms are produced through the reaction of the solvated electron with protons.
*e*_aq_*^−^* + H^+^ → H• (14)

The molecular products H_2_ and H_2_O_2_ are formed through the reactions.
H^●^ + H^●^ → H_2_
(15)
2*e*_aq_^−^ + 2H_2_O → H_2_ + 2OH^−^(16)
HO^●^ + HO^●^ → H_2_O_2_(17)

A considerable fraction of radicals formed are reconverted to water.
H^●^ + HO^●^ → H_2_O (18)
H_3_O^+^ + OH^−^ → 2H_2_O(19)
*e*_aq_^−^ + HO^●^ → OH^−^(20)

In the spur, hydrated electrons, hydrogen atoms, and hydroxyl radicals are generated in proximity. Some of these species can react with other substances, leading to the regeneration of water or the formation of molecular products such as H_2_ and H_2_O_2_. The remaining species diffuse into the surrounding solution. It takes approximately 10^−7^ s for the spur expansion to be completed. Therefore, within the nanosecond timescale, the various processes result in the production of hydrated electrons (*e*_aq_^−^), hydroxyl radicals (HO^●^), hydrogen atoms (H^●^), as well as the molecular products H_2_ and H_2_O_2_. These species are referred to as primary species, although this term does not imply that they are the first entities to be formed [91].
(21)$$H_{2}O\overset{\;\mathrm{Radiation}\;}{\to }{e_{\mathrm{aq}}}^{-},\;{\mathrm{HO}}^{•},\;H^{•},\;H_{2}O_{2},\;H_{2},\;H^{+},\;{\mathrm{OH}}^{-}$$

The hydrated electrons and hydrogen atoms are powerful reducing agents, while the hydroxyl radical is a powerful oxidizing agent. These species readily react with solutes present in the system at low concentrations. 

### 1.25. Properties of the Primary Species: The Hydrated Electron

The hydrated electron is one that is surrounded by a small number of orientated water molecules. A hydrated electron acts as a single charge anion and is a powerful reducing agent following an electron transfer process represented by:*e*_aq_^−^ + S^n+^ → S^(n−1)+^
(22)
where n is the charge on the solute [91].

### 1.26. Hydrogen Radicals

Hydrogen atoms are formed predominately in acidic solutions through the reaction:*e*_aq_^−^ + H_3_O^+^ → H^●^ + H_2_O(23)

The hydrogen atom abstracts H from saturated organic molecules increasing unsaturation. In strongly basic solutions (pH > 10), the H atom may react with OH^−^ through the reaction:H^●^ + OH^−^ ⟷ *e*_aq_^−^ + H_2_O (24)

Hydrogen atoms are less powerful reducing agents than are hydrated electrons [91].

### 1.27. Hydroxyl Radical

Hydroxyl radicals, HO^●^, act as a strong oxidant. HO^●^ will add to the double and triple bonds, abstract H atoms from organic compounds, and readily oxidize inorganic ions.
HO^●^ + S^n^ → S^n+1^ + OH^−^
(25)

#### Trapped Species

Polymer materials contain linear macromolecules that interact with their neighbors through physical, weak physio-chemical forces, and entanglements. The mobility of a polymer chain is dependent upon the physical force of the polymer. Active species formed by irradiation polymers are trapped by the limited mobility of the polymer medium. Only binary reactions with an electron or with another free radical are possible. Trapping is pronounced in semi crystalline and crystalline polymers. These trapped species remain active post irradiation.

The presence of trapped ions in irradiated samples can lead to electrical conductivity, which gradually diminishes over time in an exponential manner. Despite the decay, this induced conductivity can remain detectable for several days or even months. Trapped ion recombination in irradiated polymers has been observed to result in thermoluminescence.

### 1.28. Irradiation of Polysaccharides

#### Scission

The irradiation of polysaccharides has been extensively studied (Al-Assaf et al.) [104]. In general, the irradiation of polysaccharides in the solid state induces the formation of radicals within the macromolecules. Primary radicals are generated from the natural moisture present within the sample. These radical species lead to scission of glycosidic bonds and reduction in the molecular weight of macromolecules. Figure 4 shows the general irradiation sequence of polysaccharides.

Upon exposure of a diluted aqueous solution of a polysaccharide to ionizing radiation, most of the radiation energy is absorbed by water. The free radical reactions are induced by indirect effect and not through the direct absorption of energy. These water-derived radicals attack polysaccharide molecules [105].

As stated earlier, the radiolysis of water yields the primary radicals *e*_aq_^−^, HO^●^, and H^●^. 

The reaction between hydroxyl radicals and low-molecular-weight carbohydrates involves the removal of carbon-bound hydrogen atoms [106]. The abstraction of hydrogen atoms occurs with similar probability at all positions, resulting in the generation of radicals situated at different carbon atoms. Figure 5 provides an illustration of the potential reactions between polysaccharides and hydroxyl radicals.

The irradiation mechanisms outlined in Figure 5 show irradiation outcomes from an oxygen-free process. Oxygen will alter the outcome of radiation. The carbon-centered radicals initially engage in an interaction with oxygen, resulting in the formation of the corresponding peroxyl radicals. These peroxyl radicals undergo a variety of reactions, resulting in chain disruptions, the opening of rings, and the formation of stable oxidation products that commonly feature carbonyl groups. Figure 6 outlines irradiation reactions carried out under oxygenated conditions.

The presence of oxygen reduces the radiation-chemical yield of degradation, G(s), for polysaccharides irradiated in a solution [102].

An example of a peroxyl radical reaction other than chain scission is shown in Figure 7 for chitosan. Peroxyl radicals located in an α position to a hydroxyl or amino group will spontaneously eliminate H_2_O^●^ and O_2_^−●^.

### 1.29. Crosslinking

Polysaccharides are known to be inclined towards radiation-induced degradation, and indeed, degradation is often observed when one looks at the way in which the molecular weight changes after irradiation [102]. Many efforts have been made by scientists in the past to design ways in which the polysaccharide chains can be crosslinked by radiation, but thus far, they have only achieved minimal success in this endeavor [107]. Radiation-induced crosslinking in the presence of an acetylene atmosphere has been employed to modify polysaccharides derived from various sources. This process results in the formation of higher molecular weight macromolecules with enhanced functionalities through crosslinking upon irradiation [108]. 

### 1.30. Post Radiation Effects

Polysaccharide irradiation in solution produces free radical reactions that are fast and form during irradiation. Radicals usually decay almost instantly when radiation is stopped. Crystalline polysaccharides will have radicals trapped in their crystalline regions. These radicals will remain for long periods of time. Post irradiation processes need to be considered in applications of radiation technology in polysaccharide processing [109].

### 1.31. Role of Ionizing Radiation in Cellulose 

The modification of cellulose using ionizing radiation has numerous applications in a wide variety of fields such as healthcare, biofuels, agriculture, etc. This section aims to provide a basic discussion around the topics of the radiation chemistry of cellulose. 

Irradiation of cellulose and cellulose derivatives may be conducted in solid-state or aqueous conditions [102,110,111,112]. In the solid-state irradiation, cellulose polymers are irradiated directly, and radicals are formed along the polymer chain. The radicals then undergo degradation and/or crosslinking reactions. The dominant reaction is dependent on reaction conditions, though in the case of cellulose, chain scissions are dominant. The scissions are a result of the breakdown of the glycosidic bonds, which reduces the molecular weight of the cellulose (hence, decreasing the viscosity of the cellulose solutions). The lifetime of the radicals may vary drastically in solid-state irradiations. Some may decay instantaneously while others may become “trapped” and not decay for months. The trapped radicals occur in the crystalline region of the cellulose and migrate to the amorphous region over time. This can result in ring opening and chain scission reactions post irradiation. 

In the aqueous state irradiation, the energy of the radiation is not directly deposited onto the polymer. Instead, most of the energy is absorbed by the aqueous medium in which the cellulose is immersed. The radiolysis of water creates a series of reactive species that then tear off hydrogens from the cellulose backbone and create radicals. Thus, the formation of radicals is an indirect consequence. The main radiolysis products are the hydrated electrons (*e*_aq_), hydrogen atoms (^●^H), and hydroxyl radicals (^●^OH). The hydroxyl radical is the most reactive among these species and most responsible for the radicals formed along the cellulose backbone. The radicals decay quickly in a series of ring openings and chain scissions. 

The radiation yields of scissions (G(s)) quantifies the number of chain break events per unit of absorbed energy [102]. While degradation is the more common reaction, crosslinking of cellulose may also occur under special conditions. 

Ionizing radiation can be used to break down cellulose into its glucose units, which in turn can be converted into useful biofuels, particularly ethanol [111]. Though there are several methods to produce ethanol from cellulose, ionizing radiation has significant potential to be one of the more practical and viable techniques. However, to decompose cellulose into its monomer components, entirely by radiation, a very high dose would be needed. At these theoretically high radiation doses, the glucose would also simultaneously decompose into gluconic and glucuronic acids which cannot be converted into ethanol. Therefore, it is more practical to combine irradiation with other processes. Specifically, there has been great success with using radiation as a non-hazardous, low-energy pretreatment. Since no chemical reagents are used, the issue of removing excess reagents is not present. 

Radiation-induced graft polymerization (RIGP) can also be used to modify cellulose surfaces [112]. Radiation grafting (as opposed to other types of grafting) can be done without the need for a catalyst or any additives. Factors that are important include the type of solvent, monomer concentration, and absorbed dose used.

### 1.32. Ionizing Radiation in the Preparation of Cellulosic Feedstocks

The utilization of cellulose fibers in commercial applications continues to grow [113]. Cellulose fibers and their polymer composites offer considerable toughness, flexibility, ease of processing, recyclability, and biodegradability [114,115]. Irradiation is a useful means of modifying natural cellulose to meet different purposes [116,117]. It can be used to engineer the properties of cellulose needed to meet feedstock requirements in many areas of production. These include reinforcement polymer-based composites, cellulosic fiber production, food applications, enhancement of enzymatic hydrolysis in biofuel production, production of hydrogels, and the production of nanocrystalline cellulose [114,118,119,120,121,122,123,124,125]. 

## 2. Techniques to Identify Irradiation Effects on Cellulose

While the chemical properties and structure of cellulose have been studied for over 100 years, its ultrastructure has yet to be completely understood [126]. There are a limited number of methods available to study the cell wall structure in its natural unmodified state [127]. Most analytical methods require cellulose to be isolated and purified before it can be studied. These methods normally involve thermochemical treatments that may result in the modification of the cellulose native structure. Reviews of research into the structure of cellulose have been conducted by Hon, Zhao, Nishiyama, Eichhorn, Agarwal, Trache, Rongpipi, and Toumpanaki [31,40,128,129,130,131,132,133]. Various analytical methods have been employed to investigate the arrangement of cellulose microfibrils within cell walls [33,37,134,135,136]. These methods explore the chemical structure of cellulose, which comprises interconnected glucose units linked by β-1,4-glycosidic bonds. The microfibrils, resembling threads, are formed through the combination of glucan chains, and are stabilized by hydrogen bonding and van der Waals forces. There have been many reviews of the spectral analysis of cellulose, including O’Sullivan, Kim, and Atalla [33,37,136]. A complete understanding of the microfibril structure over multiple lengths has yet to be determined [132]. It is believed that cellulose microfibrils within cell walls have regions where the cellulose is highly ordered or crystalline while other regions are disordered and thus amorphous [130]. This hypothesis was formed from studies of samples that had been dried. Analysis of native, never-dried wood shows minimal crystalline characteristics [31]. Other studies have noticed a conjoining of microfibrils during pulping [137].

Hydrogen bonding of cellulose chains within that microfibril, including intra and inter-chain bonding, plays a role in determining cellulose’s bioactivity. There are structural differences that depend on the source of the cellulose. Structural differences are present depending upon the species from which the cellulose was obtained, and the method used in its isolation [126]. Regardless of the source of the cellulose, there are characteristic differences in 1H and 13C Nuclear Magnetic Resonance (NMR) spectra, Fourier Transform Infrared (FT-IR) spectra, and X-ray Diffraction (XRD) Analysis [31,126,136,138,139].

### 2.1. Electron Paramagnetic Resonance (EPR)

The irradiation of cellulose leads to the formation of free radicals that can be identified by EPR. EPR measurements give insight into the mechanisms of the reactions and chain scission processes that lead to the reduction of the molecular weight of the cellulose. The radicals produced after the irradiation of cellulose can be identified by calculating the g-factor of the sample. At a given frequency, the g-factor, which is independent of frequency, can be used as a fingerprint for the molecule. For example, C-centered radicals, such as the ones produced after irradiation of cellulose, have a g-factor close to the g-factor of the free electron (g ~2.0023) [140]. The EPR spectra provides more information about the sample in terms of the hyperfine interactions, which are the interactions between the magnetic moments of the electron and the nuclei of the molecules. The number of lines in the EPR spectrum, or total number of hyperfine splitting, can be calculated with the following formula: (26)$$\prod_{i}\left(2M_{i}I_{i}+1\right)$$
where M is the number of the adjacent H atoms, and I is the spin number of the nuclei [141]. The concentration of radicals in each sample can also be determined by using a standard sample with a known number of spins per gram and calculating the concentration of the sample being studied by proportion.

Studies of the radicals formed on polysaccharides have been conducted since before 1977 [142,143,144,145,146,147]. The many radicals produced in polysaccharides and the numerous reactions that these can participate in makes them difficult to properly identify. Mainly, after being exposed to ionizing radiation, excited species are formed on the polymer chain, resulting in chain scission, mostly of C-H bonds. These chain scissions result it the production of carbonyl and carboxy groups. This process is observed for most polysaccharides and its derivatives. 

The presence of a wide variety of radicals after irradiation of cellulose has been broadly reported [148,149]. Wach et al. identified 20 different radicals species after irradiation of cellulose at a dose of 10 kGy followed by exposure to higher temperatures of up to 293 K [150]. These were categorized as primary (Figure 8) and secondary radicals, and their formation was deemed to depend on different parameters, including the origin of the sample, irradiation conditions, and previous chemical and mechanical treatment of the sample [151]. It is important to mention that these primary radicals undergo further reactions between themselves, which leads to the formation of additional radicals [152,153]. 

It has been determined that the typical spectrum of irradiated cellulose is characterized by one triplet and one doublet signal [139,148,150,154,155,156,157,158]. An example of the spectrum of cellulose irradiated at different doses, from 0 to 1200 kGy, is shown in Figure 9. The triplet shows an intensity distribution 1:2:1 and a splitting of 3.0±0.2 mT. The doublet signal has a splitting constant of 2.6±0.1 mT. These signals originate from the interactions between the generated radicals on the cellulose structures and various β protons. 

This shape of the irradiated cellulose spectrum can be explained by the following. Interactions of a C-centered radical with one proton result in a doublet; according to Equation (1), the number of hyperfine splitting is calculated as $\left(2\left(\frac{1}{2}\right)\left(1\right)+1\right)=2$. In the case of interactions between a C-centered radical with two protons, utilizing again Equation (1): $\left(2\left(\frac{1}{2}\right)\left(2\right)+1\right)=3$, the interaction results in a triplet. Figure 10 shows the monomer unit of cellulose, with the accepted labeling of each atom of the pyranose ring. Each number represents the position of the carbon atoms. Radicals in the C1 position, formed by H-abstraction from the C1 atom of the ring, lead to the formation of the doublet signal. The triplet signal is obtained from the radicals generated at the C2, C3, or C4 positions, which have similar environments.

Changes from multiple EPR peaks to singlets are used to study the different radicals produced after cellulose irradiation and their evolution with time [154]. Even one year after irradiation and with a change in the spectra from a triplet to a singlet, the singlet signal can be used to identify the presence of radicals in the cellulose [155].

The concentration of radicals generated after cellulose irradiation is of paramount importance for any subsequent grafting onto the material. Through EPR analysis, the total concentration of free radicals and their decay can be studied [159]. Iller et al. conducted kinetic studies of the radicals generated in irradiated textile cellulose pulp and found a fast decay of the radicals during the first 12 h; these are the radicals located in reactive sites on the flexible surface of the cellulose [159]. They found that after the initial 12 h, the radical decay slows down and reaches a plateau ascribed to radical transfer, when the radicals become trapped in rigid domains. They observed a different behavior in the case of softwood cellulose, where the rapid decay, during the first 12 h, was followed by a period of slower decay, which lasted from 3 to 5 days, and evolved into the final decay plateau. This intermediate decay rate was determined to be a result of radical sites located in less rigid parts of the cellulose [159]. These results illustrate the dependence of the radical concentration on the structure of the cellulose being irradiated.

As mentioned earlier, the relationship between total irradiation dose and concentration of radicals can also be studied with EPR. EPR measurements conducted on unirradiated cellulose show no EPR signal which represents the absence of radicals at this stage [160]. In general, higher irradiation doses lead to a higher concentration of radicals. This relationship between irradiation dose and radical concentration appears to be linear up until moderate doses, while the grow rate slows down at higher doses [149,159,161]. The linearity of this relationship can be seen in the example shown in Figure 11, where Taibi et al. studied the effects of irradiation dose on flax fibers. A sharp increase in the concentration of radicals has been observed at doses of up to 150 kGy, after which the concentration remains constant [148].

Even radicals generated after the irradiation of cellulose at low dose and after several weeks of storage can be detected by EPR [149,161,163]. As the radicals have time to participate in different reactions, the intensity and shape of the EPR signal is altered [149]. The stability of these radicals depends strongly on the storage conditions. High temperature, humidity, and the presence of oxygen can lead to radical decay by recombination or formation of new species and result in a decrease of the EPR signal [155,156].

EPR spectroscopy is also useful when studying irradiated foods with high cellulose content such as root and leafy vegetables and some fruits. Mainly, food is irradiated to improve its shelf life and to disinfect it from insects and pests. In the US, the Food and Drug Administration (FDA) sets the irradiation doses at which each product should be irradiated [164]. Analysis of these food samples via EPR is then used to determine if it was irradiated and if it was done at the correct dose [164,165,166,167,168]. Recently, Tonyali and Yucel presented a method in which the EPR peaks of irradiated foods can be enhanced by deconvoluting the spectra [165]. Application of this method led to more in-depth analysis of the EPR peaks and better sensitivity to the irradiation dose to which the sample was subjected.

The use of EPR to study the free radicals generated in cellulose after irradiation keeps finding new applications. It has been applied to study the possible application of cotton fibers as a dosimeter during accidental radiation exposure, to investigate the effects of UV light in drugs and their active substances, and to explore the antioxidant and oxidative stability of natural foods [169,170,171,172,173]. 

### 2.2. Gas Permeation Chromatography (GPC)

GPC is commonly used to analyze the molecular weight distribution (MWD) of cellulose before and after irradiation. It has been shown that GPC coupled with a multi-angle laser light scattering detector (GPC-MALLS) provides more accurate measurements of the cellulose molecular weights and their distribution [108]. This technique gives information about the number average molecular weight (*Mn*) and the weight average molecular weight (*Mw*) in the form of a distribution. The *Mn* represents the statistical average molecular weight of all the polymer chains of the sample, while the *Mw* takes into account the contribution of the molecular weight of a chain for the molecular weight average calculation. To measure the broadness of the molecular weight distribution of the polymer, the polydispersity index (PDI=Mw/Mn) can be calculated. The degree of polymerization (DP) can also be determined with this measurement ($DP=Mw/M_{0})$, where $M_{0}$ is the molecular weight of the monomer.

In general, irradiation of cellulose leads to depolymerization and a decrease in the molecular weight of the polymer [139,159,174,175]. For example, irradiation of bamboo dissolving pulps at doses between 0 to 35 kGy resulted in a decrease of both the *Mw* and *Mn* of the polymer [176]. However, the *Mw* decreased at a faster rate, which is attributed to there being more chance of degradation in long molecular chains at low irradiation doses. Overall, the change in *Mw* is consistent with an exponential relationship between the *M_W_* and the irradiation dose. As the cellulose chains are fragmented, there are not enough electrons to continue the degradation of each chain, which slows down the process [118].

This behavior is similar at higher irradiation doses; there is a rapid decrease in the *Mw* of the cellulose due to cleavage of its polymeric chains [139]. As the sample is irradiated at high doses and in air, oxidation-degradation of the cellulose occurs, and this leads to further degradation caused by the initial irradiation. The differences between the *Mw* and the *Mn*, which are more pronounced at lower doses, can be seen in the PDI calculation. The PDI tends to decrease with increasing doses [118,159,176]. This shows that irradiation of the cellulose leads to homogenization of the molecules and a decrease in their particle size. Similarly, the DP decreases sharply when samples are irradiated at low to moderate doses (around 0–400 kGy), but becomes more moderate at higher doses (approximately 400 to 1200 kGy) [139].

Figure 12 shows the molecular weight distribution of irradiated kraft pulp at different irradiation doses. It is observed that at low irradiation doses, there is a bimodal distribution, which is attributed to the presence of high-molecular hemicellulose regions in the sample. At higher irradiation doses, the degradation increases and the bimodal distribution merges into a single distribution [118,177].

### 2.3. X-ray Diffraction (XRD)

The change in crystallinity of the cellulose after irradiation can be evaluated through XRD. The characteristic peaks of crystalline cellulose (cellulose I) for the for the 101, 110, and 200 planes are 14.8°, 16.6°, and 22.7°, respectively [139,178]. These peaks are used as the standard when looking at the crystalline structure of irradiated cellulose. Another parameter used to study crystallinity through XRD is the Segal’s crystallinity index ($CI=\frac{I_{002}-I_{am}}{I_{002}}\times 100$, where $I_{002}$ is the intensity of the peak at 2θ = 22.5° and $I_{am}$ is the lowest intensity at around 2θ = 18.5°, which corresponds to the amorphous domains in cellulose) [179]. XRD analysis of cellulose characteristic peaks at about 15.8°, 22.4°, and 34° shows a decrease in crystallinity as the dosage is increased. This observation indicates that the crystalline structure of the irradiated cellulose is damaged by the ionizing radiation [138,179,180].

At low irradiation doses, there are no significant configuration or crystallinity changes observed for irradiated cellulose (Figure 13) [176,179,181]. The characteristic peaks continue to appear at the previously mentioned locations, which is attributed to the cellulose maintaining its original crystal structure, even after irradiation. No obvious changes in the crystallinity and CI of cellulose irradiated up to doses of 60 kGy were observed [181]. 

The effects of irradiation on the crystallinity of the cellulose are more significant at higher absorbed doses. At doses over 100 kGy, the CI starts decreasing quickly as a function of dose [138,139,179]. A decrease in the CI from 75 to 58% was observed by Liu et al. for cellulose irradiated at 1200 kGy [139]. This is caused by radiation damage to the crystalline structure of cellulose. The crystallinity degradation has been attributed to the fragmentation of hydrogen bonds in the molecule of the cellulose. It is important to further investigate these changes, as decreasing the crystallinity of the cellulose makes it more accessible to chemical reagents and thus more favorable for further processing [138,176].

An article by French presented an alternative way to choose the peaks used to analyze the crystal structure of cellulose through peak deconvolution, by reducing the arbitrary degree in which they are chosen [182]. In his work, the powder diffraction patterns of the allomorphs (Iα, Iβ) and polymorphs (II, III_I_, III_II_) of the crystalline cellulose structure were calculated. His results take into account the contribution from each reflection to the peaks and present the Miller indices for each peak and an *a* axis shorter than *b* where the *a*-parameter is shorter than the *b*-parameter. 

### 2.4. Solid-State Nuclear Magnetic Resonance (ss-NMR)

ss-NMR is another important tool in the study of the structure of cellulose. It is a non-destructive technique that provides information about the chemistry, the chemical environment, and ultrastructural details of the studied material [126,183,184,185,186]. 

Starting with the 1D ^13^C cross-polarization magic angle spinning (CP/MAS) NMR technique, it was used to establish the cellulose allomorphs (Iα, Iβ) and relate the sources of the cellulose to the proportions of the crystalline structure [187]. Ren et al. recently reported the ratio of Iα to Iβ cellulose in bamboo fibers and parenchyma cells by NMR spectra analysis. In an NMR spectrum, the regions from 86 to 92 ppm correspond to crystalline cellulose, and the regions from 80 to 86 ppm correspond to the non-crystalline cellulose [126,188]. After performing and NMR analysis of the C-4 region of cellulose (identified as that from 80 to 92 ppm), the CrI was calculated and found to be between 32 and 34% for bamboo fibers and between 33 and 35% for parenchyma cells [188].

Spin-diffusion ss-NMR is another technique key in determining the inter- and intra-macromolecular distances in the cellulose structure [189]. The microfibril structure of bacterial cellulose was studied by Masuda et al. through 13 C and 1H spin diffusion ss-NMR [190]. They reported that the distance between amorphous C4 and crystalline C4 is less than 1 nm. Both the 1D and 2D spectra are shown in Figure 14. 

One of the main issues with conventional ss-NMR techniques may be insufficient resolution, which limits the understanding of the structure of low crystallinity cellulose. Kirui et al. combined a dynamic nuclear polarization (DNP) ss-NMR spectroscopy technique with chemical shift analysis to better investigate the structure of unlabeled cellulose samples [191]. They conducted the experiments by monitoring the structure of the cellulose during ball-milling. This technique allowed them to obtain high-resolution spectra of the cotton cellulose structure and to monitor the changes in Iα and Iβ allomorphs without isotope-labeling. MAS-DNP ss-NMR has also been applied to the study of the hyperpolarization across the cell wall material and to determine the structural changes in CNC conjugates caused by a response to external stimuli [192,193].

### 2.5. ^1^H and ^13^C Nuclear Magnetic Resonance (NMR) Spectra

As cellulose has limited solubility, cross-polarization/magic angle spinning (CP/MAS) solid state ^1^H and ^13^C Nuclear Magnetic Resonance (NMR) is the method of choice for studying cellulose [126,136,194,195]. There are changes in the ^1^H spectra of samples that have been irradiated with increasing dosages (0 to 1200 kGy) of ionizing radiation. The total intensity of signals significantly changes with increasing dosage. The high peak of H nuclei around 5 ppm increases and broadens. The peak also shows a shift to the left through 6 ppm. These changes are attributed to an increase of hydrogen nuclei during irradiation and resulting degradation [129,139,180].

The ^13^C CP/MAS NMR spectra show peaks for the carbon atoms of cellulose as follows: the C_1_ around 108 ppm, C_4_ between 80 and 95 ppm, and C6 around 65 ppm. These peaks grow in intensity and slowly shift to the left with increasing dosage. These slight changes in carbon chemical shifts with increasing dosage are the result of changes in the crystallinity of the cellulose and signaling degradation of the crystalline structure, and the formation of a more amorphous composition [37,136,139,180,196]. The combination of ^1^H and ^13^C-NMR spectra signifies that there is degradation in cellulose as it is subjected to high irradiation doses. The degradation increases as the dosage increases [179,197].

### 2.6. Fourier Transform Infrared (FT-IR) Spectra

The FT-IR spectra of cellulose samples irradiated with increasing absorbed doses show changes in four areas. The first area of interest is the peak at around 1732 cm^−1^, attributed to the carbonyl groups (C=O stretching vibration), which begins to appear at 200 kGy and increases in intensity with increasing absorbed dose [139,198,199]. The irradiation of cellulose will cause the formation of carbonyl and carboxyl groups due to oxidative degradation [179,200]. The second region includes the peak at around 3300 cm^−1^, attributed to the vibration of hydrogen-bonded OH-groups [139,198,199]. This peak shifts first to a lower wavenumber with doses less than 200 kGy, and shifts to higher wavenumbers when the absorbed dose exceeds 400 kGy [139,198]. At lower doses, the irradiation disrupts the intra-molecular and inter-molecular hydrogen bonding [199,201]. This will cause the 3300 cm^−1^ peak to shift to a lower wavenumber. At increased irradiation doses, the formation of carbonyl groups takes place, which produces hydrogen bonding between the carbonyl and the hydroxyl groups, resulting in a shift to high wavenumbers [199,201]. The third area of interest is the vibration wavenumbers around 1164 cm^−1^, 1112 cm^−1^, and 1058 cm^−1^ all shift to lower wavenumbers and the intensities of these bands increase with the higher irradiation dose. This is attributed to disruption of the intermolecular C-O-C bond of cellulose due to oxidative degradation [139,181]. Peaks around 1372 cm^−1^ ascribed to -C-C- and around 1237 cm^−1^ ascribed to C-O- increase with increasing dosage, indicating structural changes to the polymer backbone of cellulose [180].

### 2.7. Raman Spectroscopy

Raman spectroscopy is another important technique for studying both the amorphous and crystalline regions of cellulose [202,203,204]. The crystallinity of cellulose has been studied via Raman since 2005, and since then, various methods have been developed to better understand it [205].

Agarwal et al. proposed a method to estimate the crystallinity of cellulose (CrI) based on the Raman band at 93 cm^−1^ of cellulose I materials [204]. Their method was developed by calculating the CrI of various cellulose samples using the ratio of the height of the Raman peaks at 93 and 1096 cm^−1^ (See Figure 15). They found excellent linear correlation and were able to use this method to look at how the crystallinity of the cellulose varied at different temperatures. 

## 3. Conclusions

Cellulose is a widely used biopolymer with applications ranging from being used as a material to manufacture paper and textiles to pharmaceutical and biofuel utilization. Its biodegradability also makes it an environmentally friendly polymer with a smaller carbon footprint. Controlled use of ionizing radiation may be used to develop novel materials or processes from cellulose and its derivative.

We have discussed the various sources of cellulose found (wood, plant, bacterial based etc.) as well as its isolation and recovery. We have also covered different morphological forms cellulose may take. A brief introduction to fundamental principles of ionizing radiation was also introduced, followed by a discussion of the various radiation-induced modifications of cellulose. Some techniques to identify the radicals produced during and after irradiation were covered.

Due to the presence of the glycosidic bonds in the net cellulose and polysaccharides, ionizing radiation induces scissions on their backbone, leading to the formation of alkoxyl radicals and C-centered radicals. Hence, it is not advisable at all to irradiate cellulose in any form to enhance its mechanical properties. In addition, due to the nature of the radiolytically produced cellulose free radicals, vinyl monomers cannot add to these free radicals. As a result, these vinyl monomers cannot form covalent bonds on the backbone of the cellulose chain. Hence, no radiation-induced grafting can be achieved. As demonstrated by numerous publications, EPR has become a very powerful tool to determine the structure of radiolytically produced free radicals. By utilizing EPR spectroscopy, the oxidizing alkoxyl radicals and the C-centered radicals formed by irradiating cellulose were identified. Moreover, the reaction mechanism and kinetics of these free radicals can be elucidated and followed using EPR spectroscopy. 

Similar sessions can also occur in the radiolysis of cellulose aqueous solutions. The powerful HO^●^ radicals abstract H atoms or attack the glycosidic bonds of the cellulose chain, leading to the scission of the cellulose chains. 

It is concluded that ionizing radiation induces damages in the crystalline structure of the cellulose. The radiation-induced damage can be a result of direct interactions of ionizing radiation, and also produced from the radiation-induced chemical changes. It should also be mentioned that recrystallization can occur after irradiation, since the lower molecular scissioned molecular chains can undergo recrystallization. 

In the final analysis, ionizing radiation can effectively be utilized to recycle cellulose and its derivatives by controlling the number average degree of polymerization through radiolytic scission.

## Figures and Tables

**Figure 1 polymers-15-04483-f001:**
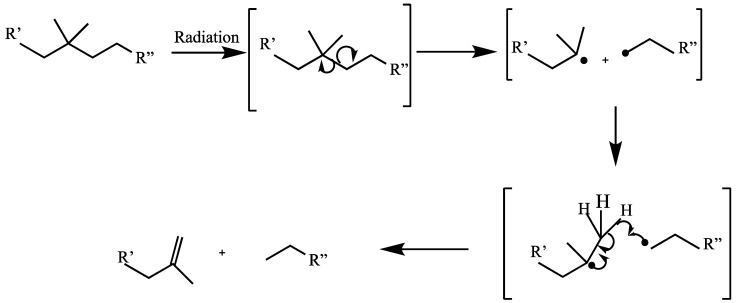
Schematic of the scissioning of an organic molecule after electron beam irradiation. The schematic shows the formation of two radicals, represented by dots, and the reactions that lead to radical decay. Figure taken from Chaudhary et al. [91].

**Figure 2 polymers-15-04483-f002:**
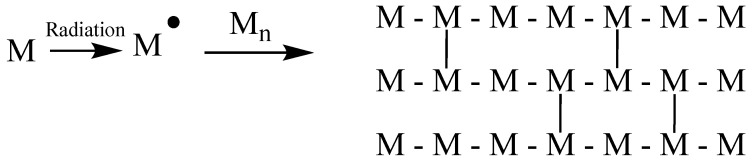
Schematic of the radiation-induced curing of a monomer (M) through polymerization and crosslinking reactions. Figure taken from Chaudhary et al. [91].

**Figure 3 polymers-15-04483-f003:**
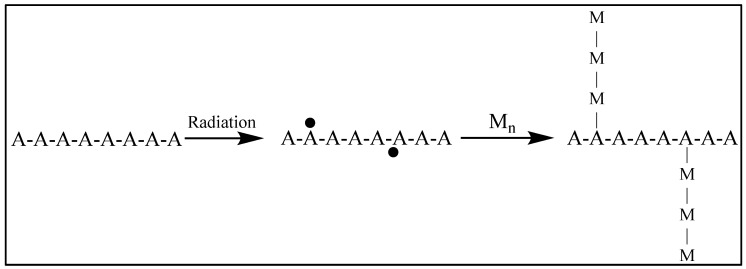
Schematic of radiation-induced grafting of monomer (M) onto polymer (A). Figure taken from Chaudhary et al. [91].

**Figure 4 polymers-15-04483-f004:**
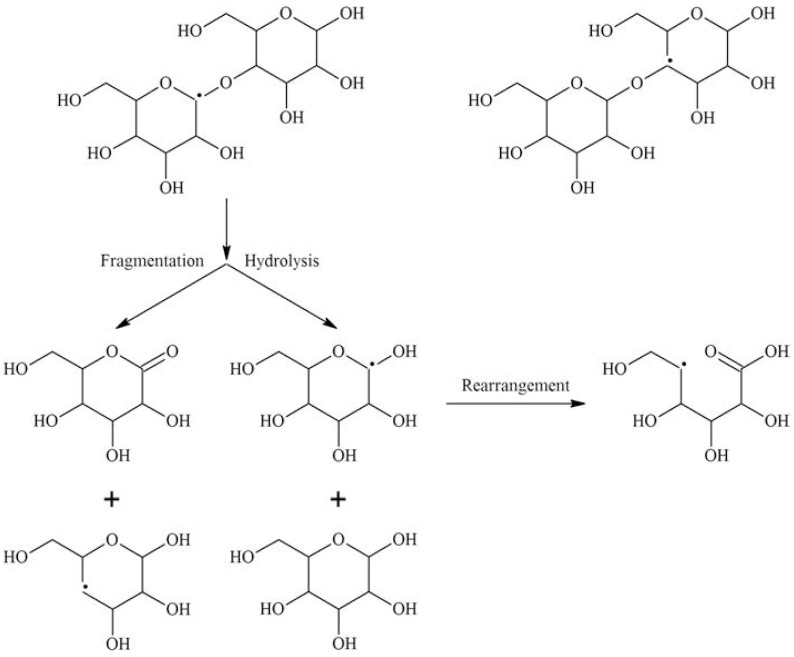
Schematic of the cleavage of the glycosidic bond and the chain scission of cellobiose during solid-state irradiation. Figure taken from Al-Assaf et al. [104].

**Figure 5 polymers-15-04483-f005:**
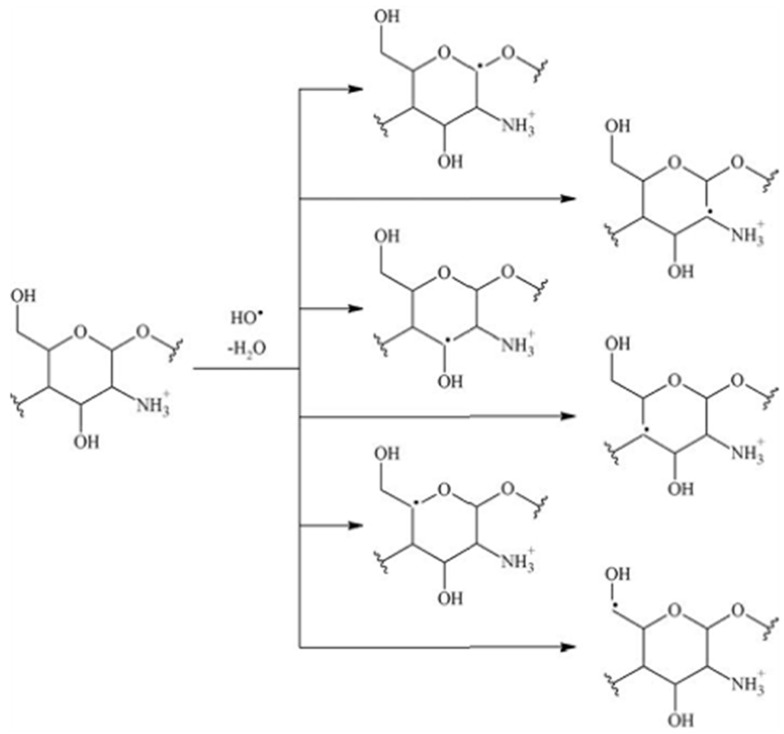
Chemical structures of radicals formed upon HO^●^ attack on a polysaccharide molecule [104].

**Figure 6 polymers-15-04483-f006:**
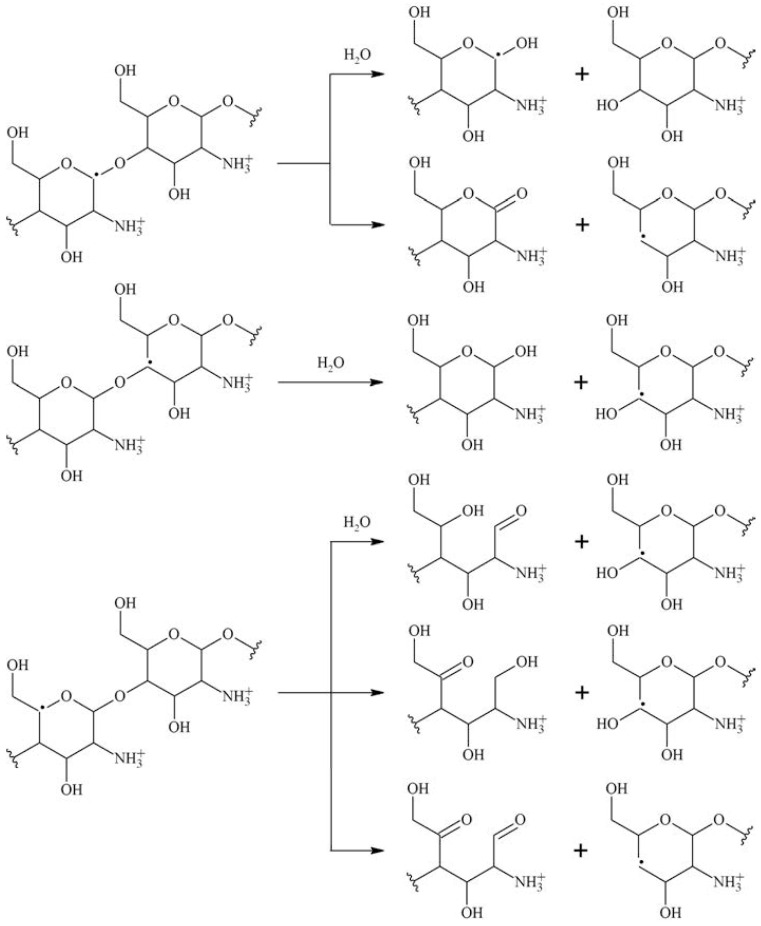
Rearrangement, hydrolysis, and fragmentation reactions during radiolysis of chitosan in oxygen-free aqueous solution [104].

**Figure 7 polymers-15-04483-f007:**
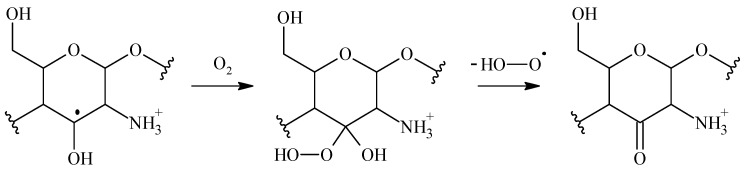
Schematic of the peroxyl radical reaction in chitosan [104].

**Figure 8 polymers-15-04483-f008:**
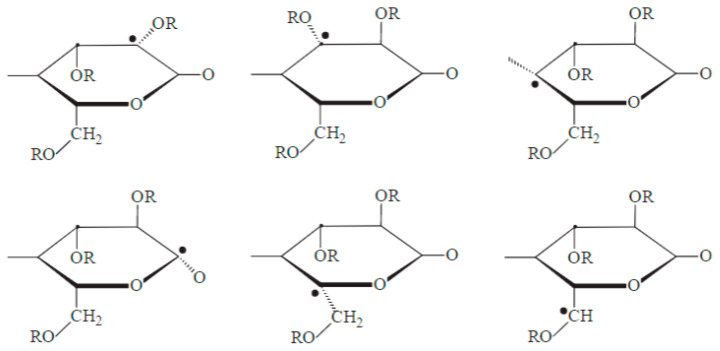
Schematic of the identified primary cellulose radicals formed after irradiation of cellulose at 10 kGy. Sample was irradiated at 77 K and heated up to 293 K. The structures were determined based on the EPR measurements, which tracked the transformation of the radicals as the temperature was increased. Figure taken from Wach et al. [150].

**Figure 9 polymers-15-04483-f009:**
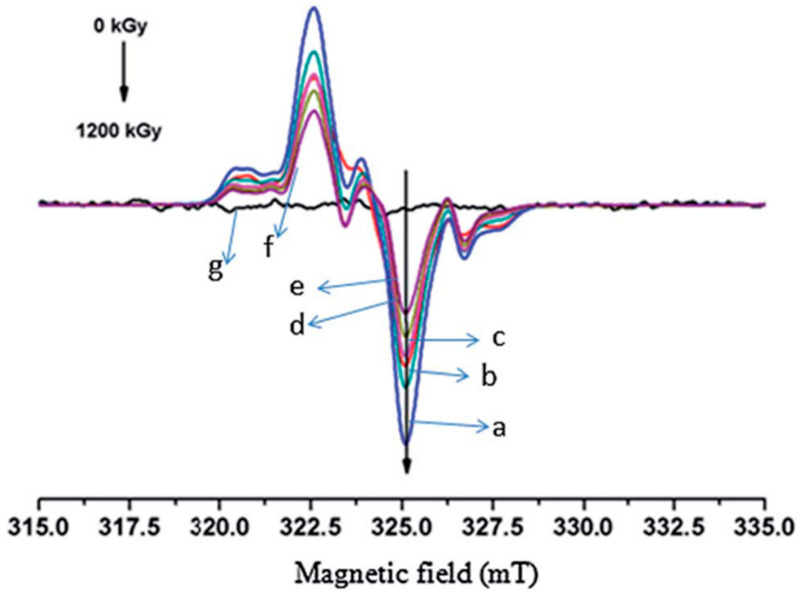
EPR spectra of irradiated microcrystalline cellulose (MCC) at different total doses (a) 1200 kGy, (b) 1000 kGy, (c) 800 kGy, (d) 600 kGy, (e) 400 kGy, (f) 200 kGy, (g) 0 kGy. The spectrum for the unirradiated (0 kGy, sample (g)) shows no radicals on the sample. As expected, as the dose is increased, a more intense signal is observed, with the sample irradiated at 1200 kGy (sample (a)) having the largest signal. Figure taken from Liu et al. [139].

**Figure 10 polymers-15-04483-f010:**
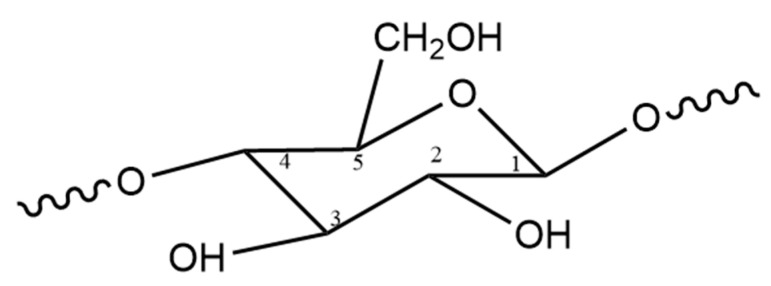
Monomer unit of cellulose with the C atoms of the pyranose ring identified according to convention. Figure is used to illustrate the origin of the signals obtained in the EPR spectrum.

**Figure 11 polymers-15-04483-f011:**
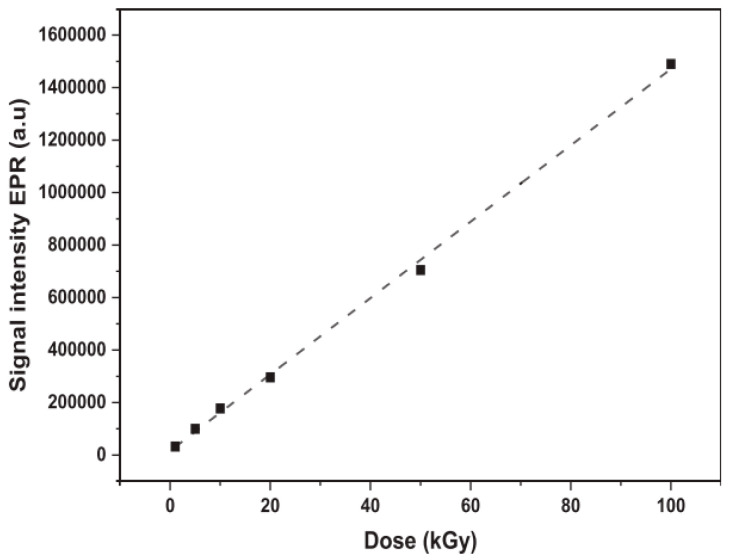
EPR signal intensity as a function of dose for flax fibers irradiated from 0 to 100 kGy. The concentration of radicals can be determined by integrating the EPR spectrum. Thus, it is directly proportional to the signal intensity shown in this figure. Figure taken from Taibi et al. [162].

**Figure 12 polymers-15-04483-f012:**
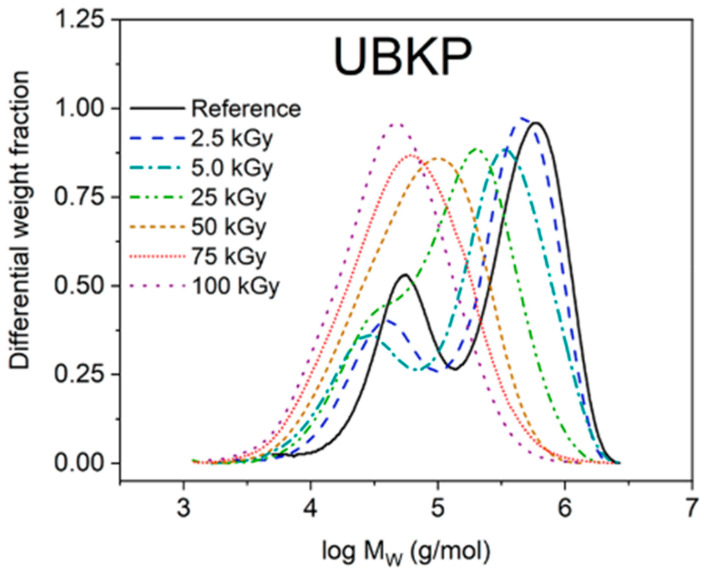
Molecular weight distribution of unbleached kraft pulp (UBKP) after electron beam irradiation at varying doses. At low does (from 2.5 to 5.0 kGy), a bimodal distribution is observed. At 25 kGy, the plot shows a transition which looks like a bump in the bell curve. After 50 kGy, there is increased degradation, which appears in the figure as a single distribution plot. Figure taken from Sarosi et al. [118].

**Figure 13 polymers-15-04483-f013:**
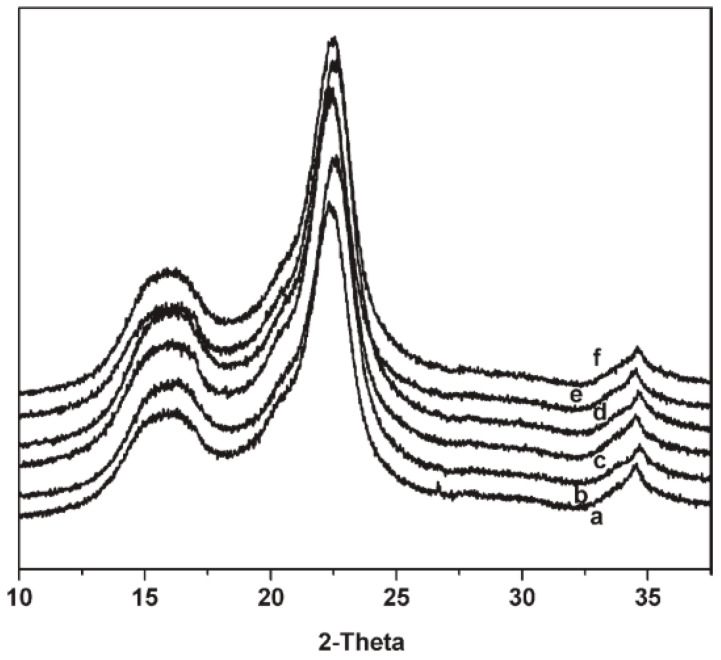
XRD spectra of irradiated cellulose at different doses: (a) 0 kGy, (b) 10 kGy, (c) 50 kGy, (d) 100 kGy, (e) 300 kGy, and (f) 500 kGy. The characteristic peaks of the cellulose at 16.6°, and 22.7° can be seen at all doses. Figure taken from Sun et al. [179].

**Figure 14 polymers-15-04483-f014:**
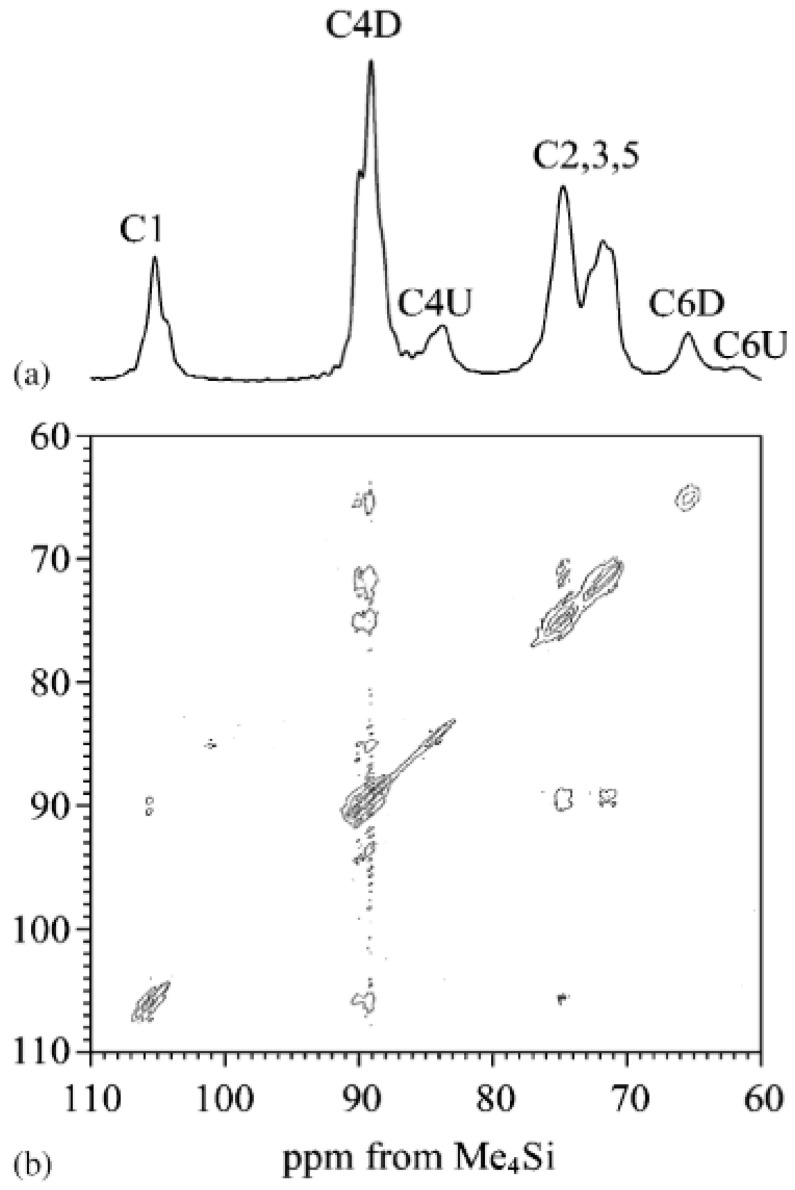
(**a**) 1D and (**b**) 2D, ^13^C NMR spectra of bacterial cellulose. The 1D spectra has been labeled with the respective C atom positions according to the cellulose structure. Figure taken from Masuda et al. [190].

**Figure 15 polymers-15-04483-f015:**
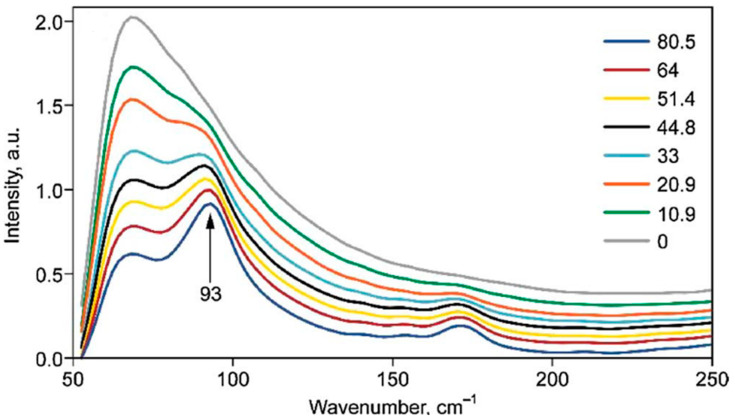
Raman spectra, normalized by the 1096 cm^−1^ peak, of the samples used for calibration with their calculated CrIs shown on the right. The samples were mixtures derived from cotton microcrystalline cellulose. Figure taken from Agarwal et al. [204].

## Data Availability

No new data were created or analyzed in this study. Data sharing is not applicable to this article.

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
