# Peer review of "On the Mechanism of the Ionizing Radiation-Induced Degradation and Recycling of Cellulose"

_polymers, 2023, doi:10.3390/polym15234483_

Round 1
Reviewer 1 Report (New Reviewer)
The purpose of this article was to review the effect of ionizing radiation on cellulose. Unfortunately, this article needs serious revision since part of the Introduction is not related to the topic of the work. In addition, the introduction contains many errors, as follows.
Line 116-117. “Microfibril bundle structures span in size from nanoscale to macroscopic dimensions with a cross-sectional dimension ranging from 2 to 20 nm”...
Remark: This statement contradicts the previously indicated diameter of elementary fibrils “The diameter of these elementary fibrils (EFs) can vary between 2 and 20 nm (lines 104-105). Yes, most researchers indicate that the diameter (lateral size) of elementary fibrils can vary between 3 and 20 nm (see e.g., Moon, R. J. et al. ref [10]); the smaller lateral size of EFs (3-4 nm) is characteristics of celluloses of herbaceous plants and wood, while the larger lateral size of EFs (15-20 nm) is characteristics of celluloses of tunicates and algae (e.g., Valonia). Regarding microfibrils (MFs) their lateral size should be larger than EFs because MFs are bundles of EFs. For example, in the cellulose of wood having EFs with the lateral size of 3-4 nm, the lateral size of MFs can be from 20 to 40 nm (see e.g., Gary Chinga-Carrasco. Cellulose fibres, nanofibrils and microfibrils.... Nanoscale Research Letters 2011, 6, 417: The diameter of MFs of Pinus rad cellulose is 28 nm, while the diameter of EFs is 3.5 nm; generally, for cellulose other plants, the diameter of MFs can be 10 times larger than that of EFs).
Lines 129-130. The thermodynamically metastable cellulose I can be converted cellulose II... Remark: The them “the thermodynamically metastable” is used incorrectly towards CI because CI cannot be converted to CII after annealing.
Remark: The fact that C1 turns into CII after chemical treatment does not mean that CI is metastable. CII can also be converted into other crystalline modifications, into CIII after treatment with liquid ammonia, and into CIV after high-temperature treatment in glycerol. Thus, the term “the thermodynamically metastable” must be removed and this sentence should be corrected, “The cellulose I can be converted into cellulose II”...
Lines 133-134. “in which the chains are arranged in a more stable antiparallel arrangement”.
Remark: Calculations of Sarco & Muggli, Woodcock, Gardner & Blackwell, and other researchers showed that there is no noticeable energy gain between antiparallel and parallel arrangements of cellulose chains. In addition, it has been proven that CI has a parallel chain arrangement. Since cellulose does not dissolve during mercerization, the parallel arrangement is preserved at the transformation of CI to CII. Thus, the phase “in which the chains are arranged in a more stable antiparallel arrangement” should be removed.
Lines 149-150. Remark: The term “particles” is unclear and should be replaced with “Cellulose morphological forms”
Line 151. Cellulose Fibers. Remark: The process of cellulose fiber formation is more complex than the authors described. First, elementary fibrils aggregate into thicker microfibrils, and then these microfibrils form layers of the cell wall and fibers as a whole.
Lines 155-157. Remark: The authors should indicate also the diameter of fibers along with their length.
Lines 161-162. “Natural filaments are produced through enzymatic peeling of fibers from wood or plant species”. Remark: According to a well-known definition, “The filament fibers refer to fibers of long continuous lengths”. The natural fibers have never been considered continuous filaments. Continuous filaments can be obtained only from melt or solution of synthetic polymers. In addition, during enzymatic treatment, fragments of plant fiber are formed instead of filaments. Thus, this sentence (in lines 161-162) should be removed as incorrect.
Line 167. “The width of CNFs ranges from 2–100 nm”... Remark: The CNFs with a width of 2 nm do not exist. The minimum lateral size of CNFs can be 10 nm (see M. Henriksson. Cellulose nanofibril networks and composites: preparation, structure, and properties. Stockholm, 2008; 51 p.).
Line 171. “Industrial production of CNCs” ... Remark: Currently, CNCs are not yet produced by industry. There is only pilot and laboratory production of CNCs. Therefore, the word “industrial” must be removed.
ETC.
I propose to remove or significantly reduce those sections of the Introduction that are devoted to cellulose, cellulose fibers, regenerated cellulose, nanocellulose, cellulose ethers & esters, application, etc., and start the Introduction with “Principles of Ionization Radiation” (line 325).
It is recommended to check English grammar.
Author Response
We would like to thank the reviewer for their feedback. To address the concerns we have done the following:
Corrections
(1) Original Line 116-117: Microfibril bundle structures span in size from nanoscale to macroscopic dimensions with a cross-sectional dimension ranging from 2 to 20 nm, depending on the source of synthesis.18,34
Revised: Microfibril bundle structures span in size from nanoscale to macroscopic dimensions with a cross-sectional dimension that vary, depending on the source of synthesis.18,34
(2) Original Line 129-132: The thermodynamically metastable cellulose I can be converted cellulose II is typically obtained by regeneration (dissolution and recrystallization) or mercerization (aqueous sodium hydroxide treatment) of native cellulose.38,39
Revised: Cellulose I can be converted cellulose II and is typically obtained by regeneration (dissolution and recrystallization) or mercerization (aqueous sodium hydroxide treatment) of native cellulose.38,39
(3) Original Line 132-133: During this conversion, the parallel chain arrangement of cellulose I changes into cellulose II, in which the chains are arranged in a more stable antiparallel arrangement.
Revised: During this conversion, the parallel chain arrangement of cellulose I changes into cellulose II.
(4) Original Line 148-149: Cellulose particles are classified by size, morphology, aspect ratio, crystallinity, and physiochemical properties. These particles are classified as follows.
Revised: Cellulose particles are classified by size, morphology, aspect ratio, crystallinity, and physiochemical properties. Cellulose morphological forms are classified as follows.
(5) Original Line 151-153: Cellulose, synthesized as long individual chains, will coalesce in a hierarchical order at the site of biosynthesis to form assemblies of elementary fibrils (protofibrils), having an approximate diameter of 3.5 nm.28
Revised: Cellulose, synthesized as long individual chains, coalesce in a series of steps forming hierarchical assemblies of elementary fibrils (protofibrils), at the site of biosynthesis with an approximate diameter of 3.5 nm.28
(6) Original Line 151-153:Cellulose, synthesized as long individual chains, coalesce in a series of steps forming hierarchical assemblies of elementary fibrils (protofibrils), at the site of biosynthesis with an approximate diameter of 3.5 nm.28 Fibrils are then clustered into cellulosic fibers that when processed produce fibers in three typical geometries, strand fibers (long fibers of 20-100 cm length), staple fibers (short fibers of 60 mm length), and pulp fibers (very short fibers of 1–10 mm length).46
No revision made line 151- 153 provides enough general information about diameter of fibers.
(7) Original Line 161-162: Natural filaments are produced through enzymatic peeling of fibers from wood or plant species
Revision: Line deleted
(8) Original Line 167: The width of CNFs ranges from 2–100 nm depending on the source of cellulose, fibrillation process, and pretreatment.18
Revision: The width of CNFs ranges from 10–100 nm depending on the source of cellulose, fibrillation process, and pretreatment.18
(9) Original Line 171: Industrial production of CNCs is achieved through sulfuric acid hydrolysis and the ultrasonic treatment of bulk cellulose
Revision: Large scale production of CNCs is achieved through sulfuric acid hydrolysis and the ultrasonic treatment of bulk cellulose.
(10) While the focus of this review article is on ionizing radiation-induced degradation and recycling of cellulose, it is important for the reader to have an understanding of the properties of the diverse feedstocks that are available.
Thank you.
Reviewer 2 Report (New Reviewer)
This paper analyzed the mechanism of ionizing radiation-induced degradation and cellulose.
How about the effect of cellulose natural structure and distinction on the radiation-induced degradation of cellulose.
Maybe the source, or functionalization of cellulose can be concluded in one table or figure to highlight this section.
More relevant reference can be cited, such as https://doi.org/10.1016/j.cej.2022.139786
The language is acceptable
Author Response
We would like to thank the reviewer for their comments. We greatly value the feedback. Here are our responses:
Reviewer Comment: How about the effect of cellulose natural structure and distinction on the radiation-induced degradation of cellulose.
Response:
We talk about chain scissions which are induced on the backbone of the polymer chains starting on line 395. The scissions are what leads to the degradation of cellulose via ionizing radiation. We have revised the introductory sentence to make this clearer:
Revised: Chain scissioning is the breaking of chemical bonds on polymeric chains which leads to the degradation of the cellulose. This results in a reduction in the average size or molecular weight of the macromolecule and is different from, depolymerization, which results in in the release of the original monomer.
Reviewer Comment: More relevant reference can be cited, such as https://doi.org/10.1016/j.cej.2022.139786
Response:
We have read the reference cited by the reviewer and while we find the work very interesting, we do not believe it would fit under the ionizing radiation category that the paper is covering.
We would thank the reviewer again for their feedback. Thank you.
Round 2
Reviewer 1 Report (New Reviewer)
Version 2 of the manuscript contains some shortcomings and needs additional correction.
Line 148. Cellulose particles. Remark: The term “particles” should be replaced with the term “morphological forms” because cellulose fibers, filaments, nano-fibrils, and regenerated cellulose are not “particles”
Lines 153-154. Fibrils are then clustered into cellulosic fibers that when processed produce fibers... Remark: Here is an error since the term “fibers” is repeated twice. In addition, “fibers” cannot produce “fibers”. Instead, the authors should write, ”Elementary fibrils are then clustered into cellulosic microfibrils that produce fibers”...
Line 300. Cellulose application. Remark: Since in this section also cellulose derivatives are mentioned, the name of this section should be “Application of cellulose and its derivatives”.
Lines 330, 333, 334, 347, and 361. Physical phase; Physico-chemical phase; Chemical phase; phase. Remark: The term “phase” is not appropriate here. Replace it with the term “stage”, as follows, Physical stage; Physico-chemical stage; Chemical stage; in line 361, replace “phase” with “stage”.
Lines 334-336. Remark: This sentence contains errors and should be corrected, as follows, “The physical stage occurs within 10–13 s after exposure to ionization radiation when electrons are knocked out of the target material as a result of which positive ions are created in this material”
Lines 370-371. crosslinking or polymerization, Remark: Put a comma after “crosslinking” and remove “or”, because “crosslinking” is not the same as “polymerization” process
Line 374. Remark: Instead of “Crosslinking”, write “Crosslinking and polymerization”
Lines 375-377. Remark: The definition “crosslinking” was given incorrect and must be corrected, as follows, “Cross-linking is the process of forming transverse covalent bonds between polymer chains to form a three-dimensional structure”
Line 378. carbon hydrogen bonds Remark: Correct this phrase as follows, “bonds between carbon and hydrogen in neighboring polymer chains”...
Line 391-394. Remark: This fragment contains scientific errors and should be corrected, as follows, “Unsaturated compounds, especially vinyl monomers, undergo efficient polymerization when exposed to ionizing radiation. During this process, the double bonds of monomers are broken, which leads to a chain elongation reaction with the formation of a polymer”.
Lines 396-402. Remark: This fragment contains scientific errors and should be corrected, as follows, “Chain scission is the breaking of the chemical bonds of polymer chains, resulting in degradation of the polymer, e.g., cellulose. This leads to a decrease in the length and molecular weight of the macromolecule. Radicals formed in polymer materials with side branches transfer to hydrogen atoms forming a double bond; in this case, the scission becomes permanent. Figure 1 illustrates the process of permanent main-chain scission.103
The cleavage of some linear polymers, e.g., polyolefins, tends to be restored due to the recombination of neighboring radicals”.
Line 417. reactive liquid Remark: Write, “reactive liquid monomer”
Line 431. in a chemically inert atmosphere Remark: Write, “in a vacuum or inert gas atmosphere” ....
Line 437. some homo polymerization of the monomer Remark: Remove “some” and write, “graft-polymerization of the added monomer”
Lines 439-440. - Ionizing radiation will produce radical, ionic, and mixed radical – ionic mechanisms Remark: Remove the hyphen (-) before Ionizing and correct this sentence as follows, “Ionizing radiation will produce radicals, ions, and combination of radicals with ions”.
Line 443. initiate polymerization Remark: Write, “initiate graft-polymerization of added monomer”
Line 452. efficiently, resulting in the generation Remark: Write, “efficiently, but also to the generation”..
Line 458. in all polymer irradiation processes Remark: Write, “in irradiation process of polymers”
Line 463. side-branches Remark: Remove the hyphen, i.e., “side branches”
Lines 461-462. The yield of hydrogen ... is 4 to 1. Remark: The authors should clarify, whether was it the ratio of mass parts, molar ratio, or yield expressed in a ratio of volume H2 to a mass of the polymer.
Lines 845-846. The characteristic peaks of crystalline cellulose (cellulose I) for the 101, 10-1 and 002 lattices are 14.8°, 16.6°, and 22.7°, respectively. Remark: The authors used the old indexation of Mayer & Mish early 20th century. Therefore, these old indices, for the 101, 10-1, and 002 lattices, must be replaced with modern Miller indices, as follows, “for the 1
0, 110, and 200 planes”...
Line 848. crystallinity index (CI...) Remark: Write, “Segal’s crystallinity index (CI...)”
Line 850. which corresponds to the amorphous content of crystalline cellulose Remark: This phrase must be corrected, “which corresponds to the amorphous domains in cellulose”
Line 851. at about 15.8, 22.4, and 34 lattices Remark: Remove the word “lattices”, i.e., at about 15.8, 22.4, and 34.
Lines 857-859 No obvious changes in the crystallinity and CI of cellulose irradiated up to doses of 60 kGy proves that high energy radiation can penetrate both the crystal and amorphous regions of the cellulose.
Remark: Here is the logical error: if high energy radiation can penetrate both the crystal and amorphous regions then the crystalline structure can be changed. Therefore, this sentence should be corrected, as follows, leave only the beginning of the sentence “No obvious changes in the crystallinity and CI of cellulose irradiated up to doses of 60 kGy”. And the end of the sentence, “proves that high energy radiation can penetrate both the crystal and amorphous regions of the cellulose”, needs to be deleted.
Lines 867-868. This represents the formation of amorphous cellulose and damage to the crystalline structure. Remark: This sentence should be corrected, as follows, “This is caused by radiation damage to the crystalline structure of cellulose.”
Lines 877-878. ...conforming to the c fiber axis, a righthand relationship between the axes, and an a axis shorter than b. Remark: This phrase should be corrected, as follows, (1) remove “...conforming to the c fiber axis, a righthand relationship between the axes“; (2) replace “and an a axis shorter than b“ with“where the a-parameter is shorter than the b-parameter“.
The English should be improved and corrected.
Lines 331-332. The interactions between an ionizing radiation an organic material occurs in a three step process. Remark: Correct the English as follows, “The interactions between ionizing radiation and organic material occur in a three-step process”
Line 350. to to the positive charged ions. Remark: Remove repeated “to” and write, “to the positively charged ions”.
Line 399. in in the release Remark: Remove repeated “in”
Line 414. the yield from scissioning Remark: Correct this phrase, write, “the yield for scission”
Line 448. scissioning Remark: Replace “scissioning” with “scission”
Line 470. Per oxyradicals Remark: Write, “Peroxyl radicals”
Lines 552-553. induce the formation of radicals with in Remark: Write, “induces the formation of radicals within”
Line 554. scission of glycosidic bonds is reduction Remark: Write, “to scission of glycosidic bonds and reduction”...
Line 592. radiation induced Remark: Write, “radiation-induced”...
Line 594. Much efforts Remark: Write, “Many efforts”...
Line 602. produce Remark: Write, “produces”
Line 604. Crystalline polysaccharide Remark: Write, “Crystalline polysaccharides”
Line 606. on applications of radiation technology on... Remark: Write, “in applications of radiation technology in”...
Line 607. Role of Ionizing Radiation of Cellulose Remark: Write, “Role of Ionizing Radiation in Cellulose”
Line 611. in solid state Remark: Write, “in solid-state”
Line 612. In the solid-state Remark: Remove “the” and write, “In solid-state” ...
Line 617. ( ...the viscosity of the cellulose) Remark: Write, (... the viscosity of cellulose solutions)
Line 625. than abstract hydrogens Remark: Write, “then tear off the hydrogen”...
Line 629. series of ring opening Remark: Write, “a series of ring openings”...
Line 633. under right conditions Remark: Write, “under the special conditions”
Line 641. practical combine irradiation Remark: Write, “practical to combine the irradiation”...
Line 642 low energy Remark: Write, “low-energy”
Line 652. easy of processing Remark: Write, “ease of processing”
Line 655. polymer based Remark: Write, “polymer-based”
Line 906. high resolution Remark: Write “high-resolution”
Line 964. application Remark: Write plural form, “applications”
Line 966. It's biodegrability Remark: Correct it as follows, “Its bio-degradability”
Line 972. ...were... Remark: Replace ”were” with “was”
Lines 972-973 radiation induced application Remark: Write 'radiation-induced”
Line 979. onto Remark: Replace 'onto“ with 'to”
Line 982. from numerous Remark: Write “by ‘numerous”
Line 983. the radiolytically Remark: Remove “the” and write “radiolytically”
Line 984. C-centerd Remark: Write 'C-centered”
Line 985. was identified Remark: Write 'were identified”
Line 986. and follow using EPR Remark: Write instead 'and monitored using ESR”
Line 988. abstract H atom Remark: Write instead 'tear off of H-atoms”
Line 990. damages in Remark: Write instead 'damage to“
Line 993. after the irradiation Remark: Remove 'the“ and write ' after irradiation”
Author Response
Hello,
Please see our response to the reviewer attached. We thank you for your feedback.

This manuscript is a resubmission of an earlier submission. The following is a list of the peer review reports and author responses from that submission.
Round 1
Reviewer 1 Report
The review generally provided thoughtful summary of the ionizing radiation of cellulose materials and the further degradation and recycling. I recommend that the paper can be accepted after major revision.
Abstract: The paragraph is too short to highlight the whole review paper. Some major points of ionizing radiation as well as the perspectives of this review are recommended to be included.
The part of “cellulose sources” is recommended to be replaced ahead of the “cellulose structure”. There is no new information in the part introduction of cellulose. Please cite some update studies on cellulose structure and application.
What is the relationships between the “principles of ionization radiation” and cellulose science. This should be emphasized.
No examples on cellulose reaction based on the different mechanisms were found.
The technique for determining cellulose structure by XRD is not specific. Deconvolution of crystalline and amorphous regions were used rather than the old equation. Please refer to French 2014, Ling et al. 2019….. Moreover, more techniques such as NMR and Raman were also used frequently for cellulose structure analysis.
Author Response
The review generally provided thoughtful summary of the ionizing radiation of cellulose materials and the further degradation and recycling. I recommend that the paper can be accepted after major revision.
- Abstract: The paragraph is too short to highlight the whole review paper. Some major points of ionizing radiation as well as the perspectives of this review are recommended to be included.
Our response: We have expanded on the abstract.
- The part of “cellulose sources” is recommended to be replaced ahead of the “cellulose structure”. There is no new information in the part introduction of cellulose. Please cite some update studies on cellulose structure and application.
Our response: Thank you for your observations. We agree and have rearranged the introduction section to show the “cellulose sources” before going into the “cellulose structure”. Furthermore, the introduction section was updated to include new studies on cellulose structure and application. The following section was added:
“Cellulose Application
The tunability of the cellulose structure, and thus of its physical and chemical properties, makes cellulose a leading material for advanced technologies.80 Due to the high number of possible combinations of the previously discussed parameters (i.e., source, structure, and dissolution system) which result in cellulose materials with a large variety of properties, the applications are endless. One of the current focuses of cellulose research applications is on the skin-care industry.81–83 Deposition of chitin nanofibrils on the skin through an electrospray method has been found to lead to anti-inflammatory activity. Azimi et al. found that the source and dissolution system chosen did not have any negative effects on the results.81 Recently, interesting biological activity has been found in carboxymethyl cellulose (CMC).82 CMC is commonly used as a structural element. However, it has been found that it has inflammation control properties and has been presented as a possible replacement of surfactants used in cosmetics, which makes it an ideal alternative for skin-care applications.84,85
Optoelectronic applications of cellulose have also been increasing in this last decade.86 For example, the use of cellulose as high definition displays for flexible touch screen panels is being explored. Zhu et al. fabricated a highly transparent cellulose-based paper that could be used in high-definition displays.87 The possibility of fabricating these transparent films also opened the door for using cellulose in other flexible electronic applications. Chen et al. studied the fabrication of transparent and hydrophobic cellulose films, with not only an optical transparency of up to 92.3%, but also a tensile strength of 198.7 MPa.88 These have promising applications for both electronic device protection and emerging electronics.”
- What is the relationships between the “principles of ionization radiation” and cellulose science. This should be emphasized.
Our response: We delve into the principles of ionizing radiation of polysaccharides later in the section, but have added the following to the first paragraph to emphasize its importance:
“Before delving into the effects of ionizing radiation on cellulose, this section will give an overview of the effects of ionizing radiation on matter. Since polysaccharides have various conformations, the radiation effects will depend on the participation of specific monomers present in the cellulosic chain.1 When cellulose is irradiated, free radicals are formed and interact with the solvent present, in the same way as synthetic polymers undergo free radical reactions.”
- No examples on cellulose reaction based on the different mechanisms were found.
Our Response: The mechanisms describing the irradiation of cellulose can be found in the “Irradiation of polysaccharides” section.
- The technique for determining cellulose structure by XRD is not specific. Deconvolution of crystalline and amorphous regions were used rather than the old equation. Please refer to French 2014, Ling et al. 2019…..
Our Response:
Thank you for your recommendation. The following was added to the XRD section:
“An article by French presented an alternative way to choose the peaks used to analyze the crystal structure of cellulose through peak deconvolution, by reducing the arbitrary degree in which they are chosen.139 In his work, the powder diffraction patterns of the allomorphs (Iα, Iβ) and polymorphs (II, IIII, IIIII) of the crystalline cellulose structure were calculated. His results take into account the contribution from each reflection to the peaks and present the Miller indices for each peak conforming to the c fiber axis, a right-hand relationship between the axes, and an a axis shorter than b.”
- Moreover, more techniques such as NMR and Raman were also used frequently for cellulose structure analysis.
Our Response: Thank you again for your input. We agree and have added these two techniques to the discussion of cellulose structure analysis.
Reviewer 2 Report
This paper is very well written and only needs to be revised in some points. The state of research is very well researched and documented with a large number of papers. The graphics, especially of the chemical structures, are clearly arranged.
The sentence "Polymerizable monomers are mixed with short-chain The curing is a free – radical process where oxygen has an inhibiting effect." on page 10 is not completed.
Maybe there is an typo on page 12 at the beginning of the sentence "Must biomass materials contain water". Do you mean "most"?
There are some different font types and sizes on the following figures:
-Page 19, Figure 9
-Page 20, Figure 10 & 11
-Page 22, Figure 12
-Page 23, Figure 13: also named as Figure 2
The conclusion provides a good overview about the discussed topic. Maybe an additional sentence following the "All in all, ionizing radiation has a lot to offer polymer scientists." on page 23 could bring in some more details to the potential of the field of the ionizing radiation in the context of recycling.
Author Response
This paper is very well written and only needs to be revised in some points. The state of research is very well researched and documented with a large number of papers. The graphics, especially of the chemical structures, are clearly arranged.
- The sentence "Polymerizable monomers are mixed with short-chain The curing is a free – radical process where oxygen has an inhibiting effect." on page 10 is not completed.
Our Response: Thank you for the observation. This was fixed by removing the incomplete sentence from the paragraph.
- Maybe there is an typo on page 12 at the beginning of the sentence "Must biomass materials contain water". Do you mean "most"?
Our Response: Yes – we have fixed it to “most biomass materials contain water”.
- There are some different font types and sizes on the following figures:
-Page 19, Figure 9
-Page 20, Figure 10 & 11
-Page 22, Figure 12
-Page 23, Figure 13: also named as Figure 2
Our Response: Thank you for pointing this out. We have fixed the fonts, made sure they all follow the same style, and corrected the numbering.
- The conclusion provides a good overview about the discussed topic. Maybe an additional sentence following the "All in all, ionizing radiation has a lot to offer polymer scientists." on page 23 could bring in some more details to the potential of the field of the ionizing radiation in the context of recycling.
Our Response: We have expanded on the conclusion.
Reviewer 3 Report
The paper „On the Mechanism of the Ionizing Radiation-Induced Degradation and Recycling of Cellulose“ aims to demonstrate the influence of ionizing radiation on cellulose. I do not see described effect of recycling on cellulose properties. The authors have mentioned a very large number of cellulose derivatives, but the information given are minor and only informative character. A very big part of the information presented in this paper are very well known and described already.
The paper is not prepared according to Journal instructions.
The abstract of the manuscript should emphasize the key facts that this work contributes to. The title of the paper should contain that the paper is a review paper.
The plagiarism software analysis showed very high percentage of similarities to other studies (more than 30 %) in the parts regarding Introduction, cellulose structure, and its derivatives. This is not acceptable and must be corrected.
Figure 13 is taken from someone else’s research, altogether with copied figure caption…
The conclusion given is already very well known and I do not see the potential of this paper or any new scientific contributions.
Author Response
The paper „On the Mechanism of the Ionizing Radiation-Induced Degradation and Recycling of Cellulose” aims to demonstrate the influence of ionizing radiation on cellulose. I do not see described effect of recycling on cellulose properties. The authors have mentioned a very large number of cellulose derivatives, but the information given are minor and only informative character. A very big part of the information presented in this paper are very well known and described already.
- The paper is not prepared according to Journal instructions.
Our response: You are right. We have corrected this by using the MDPI template.
- The abstract of the manuscript should emphasize the key facts that this work contributes to.
Our response: We have expanded on the abstract.
- The title of the paper should contain that the paper is a review paper:
Our response: Thank you for pointing this out. The manuscript has been edited according to the MDPI template and now includes the “Review” title.
- The plagiarism software analysis showed very high percentage of similarities to other studies (more than 30 %) in the parts regarding Introduction, cellulose structure, and its derivatives. This is not acceptable and must be corrected.
Our response: This issue has been corrected. Thank you for pointing this out.
- Figure 13 is taken from someone else’s research, altogether with copied figure caption…
Our response: Thank you for bringing this up. We have added the correct citations for all the figures used in this review paper, which have been taken from published papers.
- The conclusion given is already very well known and I do not see the potential of this paper or any new scientific contributions.
Our response: We have expanded on the conclusion.
Round 2
Reviewer 3 Report
After the review, the plagiarism is only 22 %, thus it can be concluded that the authors did not make requested changes.
The changes made by the authors are only "cosmetic". Again, the paper contains a very wide theory, but the information given are minor and only informative character. The authors did not improve it in a way that was needed.
The authors did not use the unique writing style, for example somewhere they mention irradiation but in the next sentence they use radiation. The terminology should be unique.
Description of the figures are only of informative character. There is no specific explanation. For the readers who have a limited knowledge, the paper cannot be helpful at all.
Some paragraphs are described by the use of only one reference.
The paper is not prepared according to Journal instructions.